# New Mineral Occurrences in Massive Sulfide Deposits from Mănăilă, Eastern Carpathians, Romania

**Gheorghe Damian [1],\*, Andrei Ionuț Apopei [1], Andrei Buzatu [1], Andreea Elena Maftei [2] and Floarea Damian [3]**

[1] Department of Geology, Faculty of Geography and Geology, "Alexandru Ioan Cuza" University of Iași, 700505 Iași, Romania

[2] Institute for Interdisciplinary Research, Department of Exact and Natural Sciences, "Alexandru Ioan Cuza" University of Iași, 700057 Iași, Romania

[3] Carpathian Association for Environmental and Earth Sciences, 430142 Baia Mare, Romania

\* Correspondence: gheorghe.damian@uaic.ro

**Abstract:** The massive sulfide deposits (VMS) from Mănăilă are associated with the metamorphic formations of the Tulgheș Lithogroup from the Bucovinian Nappes of the Crystalline-Mesozoic Zone in the Eastern Carpathians, Romania. The following types of ore were identified: pyrite-polymetallic, pyrite copper, compact and precompact copper, and quartz-precompact copper. The polymetallic mineralization consists of pyrite, chalcopyrite, sphalerite, galena, and subordinately arsenopyrite and tennantite. The copper, especially the quartz-copper mineralizations, have a distinct mineralogical composition compared to the other metamorphosed mineralizations of the Tulgheș Lithogroup. These types of deposits from Mănăilă contain large amounts of bornite and chalcocite along with chalcopyrite. Tennantite is abundant and has up to a 3.57 wt.% of bismuth. Wittichenite was identified for the first time in the metamorphic mineralizations and mawsonite was identified as the first occurrence in Romania. An unnamed mineral with the formula: $(Cu, Fe)_{11}(Pb, Ag)S_7$ was also identified, belonging to the sulfides group. The compact and precompact pyrite-rich ores, located in sericite $\pm$ quartzite schists and covered by rhyolitic metatuffs, are of hydrothermal-sedimentary type metamorphosed in the greenschist facies. The source of the quartz-copper mineralization would be the retromorphic or metasomatic hydrothermal solutions that circulated through major fractures.

**Keywords:** Eastern Carpathians; Tulgheș Lithogroup; Mănăilă ore deposit; VMS; Raman spectroscopy; tennantite; wittichenite; mawsonite

## 1. Introduction

The sulfide mineralizations from Mănăilă are located in the northwestern part of the Fundu Moldovei mining area in the Eastern Carpathians (Figure 1a). The Eastern Carpathians chain was formed in the Alpine tectonic cycle and consists of five zones arranged parallel along the chain, reflecting the major stages of a geosynclinal evolution [1,2]. The areas from west to east are the crystalline Mesozoic, the flysch zone, and the molasses zone. In the western part of the Eastern Carpathians, the Transcarpathian zone and the Neogene volcanics are observed. The mineralization is located in the rocks of the Tulgheș Lithogroup from the crystalline-Mesozoic zone of the Eastern Carpathians, which structurally belongs to the Eastern Dacides [3,4]. The crystalline-Mesozoic zone is made up of several alpine tectonic units, with eastern vergence, overthrusted in the Austrian tectogenesis [4]. The middle part of the Eastern Carpathians known as the "Crystalline-Mesozoic Zone" consists of a metamorphic foundation that supports the Permian-Mesozoic sedimentary cover [5].

The ore deposits associated with the low-grade metamorphic formations of the Tulgheș Lithogroup have a major interest from an economic point of view due to their significant concentrations of polymetallic sulfides [6], being exploited especially for their high contents in Cu (0.25%–2.32%), Zn (0.83%–6.30%), and Pb (0.46%–3.93%). The polymetallic sulfide

mineralization constitutes a cantoned belt in the Tg3 horizon of the Tulgheș Lithogroup [4] that stretches from the north of the Poienile de sub Munte—with some interruptions—to the south of Bălan, with a significant extension along the chain of ~200 km [6,7]. The concentration of mineralizations in a single horizon (Tg3) reveals the fact that they were subjected to stratigraphic control [6,7]. The belt of polymetallic mineralizations associated with the epimetamorphic schists of the Eastern Carpathians, with a NW-SE direction, underwent alpine fragmentation and is located in the Bucovinian nappe, the upper unit of the Central-Eastern Carpathian nappes [6,7]. The main mineralizations of the polymetallic belt accumulated in the Tulgheș Lithogroup are grouped into three districts, as follows [6]:

Borșa—Vișeu District

Fundu Moldovei—Leșul Ursului District

Bălan—Fagul Cetății District

The sulfide mineralizations of the Fundu Moldovei zone are concentrated in the Bucovinian nappe of the Tulgheș group; in the Fundu Moldovei member and only partially in the Bacșa member. The Mănăilă mineralization occurs in the upper part of the Fundu Moldovei member. The first studies on mineralization from the Tulgheș Lithogroup were carried out by [7–9]. The mineralizations are presented in the form of layers and lens-layers of variable sizes composed of a massive and disseminated finely granular ore; sometimes folded and arranged concordantly in the metamorphic formations [7,10]. The texture of the mineralization is compact with a granular structure; but in the precompact and impregnation ore, it presents as a parallel, fine shale texture [9], with superimposed cataclastic textures due to metamorphism. The mineralizations consist of pyrite and subordinate chalcopyrite; sphalerite; galena; and subordinate amounts of arsenopyrite; pyrrhotite; and bismuthinite. Sulfosalts are rare and are represented mostly by tetrahedrite.

## 2. Geological Setting

Balintoni [4] separates the metamorphic foundation of the Eastern Carpathians into five lithogroups: the Bretila Lithogroup (Middle Proterozoic), the Rebra Lithogroup (Upper Proterozoic), the Negrișoara Lithogroup (Upper Proterozoic), the Tulgheș Lithogroup (Cambrian-Lower Ordovician), and the Rodna Lithogroup (Silurian–Devonian). Within the Tulgheș Lithogroup are the most important reserves of syngenetic ores from the crystalline Eastern Carpathians. In the Tulgheș Lithogroup, four formations were separated, Tg1, Tg2, Tg3, and Tg4.

The metamorphic rocks of the Tulgheș Lithogroup are part of three Mesocretaceous tectonic units: Bucovinian and Sub-Bucovinian Nappes, Infrabucovinian Nappe, which were named by [3], and Central-Eastern Carpathian Nappes or "Median Dacides". Later, according to the terminology of [4], they were defined as "Eastern Getides".

According to the research of Munteanu and Tatu [11], the Tulgheș Lithogroup would be an island arc connected to the eastern part of the Avalonian continent detached from Gondwana in the Lower Ordovician and overlain in the Eastern European craton at the end of the Ordovician-Lower Silurian.

The Tulgheș Lithogroup, in which the massive sulfide polymetallic mineralizations from the Fundu Moldovei area are located, is separated into five lithostratigraphic units (formations Tg1—Tg5) [10]:

Tg1—blastodetritic-quartz formation;

Tg2—predominantly graphitic formation with black quartzites;

Tg3—metasedimentary rhyolitic formation, associated with polymetallic mineralization;

Tg4—blastodetritic-phyllitic formation;

Tg5—predominantly graphitic formation, with green shale and limestone.

Vodă and Vodă [12] and Balintoni [4] divided the lithogroup into four lithostratigraphic units called "lithozones"—Căboaia, Holdița, Leșul Ursului, Arșita Rea. Later, the Tulgheș Lithogroup was divided by some authors into four formations [13,14] as follows:

Tg1—quartzite and sericite-chlorite schists;

Tg2—sericite-chlorite schists ± organic matter, Mn ore, limestone, and carbonate schists;

Tg3—meta-rhyolites ± Cu and metal sulfide deposits, sericite-chlorite schists, and greenschists;

Tg4—sericite-chlorite schists, metagraywacke, meta-rhyolite, greenschists, and limestone.

The central part of the Tulgheș Lithogroup is a volcanogenic–sedimentary association consisting predominantly of acid metavolcanites (porphyrogenic rocks, Figure 1) with considerable quantitative variability from a mineralogical and chemical point of view [4], developed in a basin near an island arc. The Tulgheș Lithogroup is of epimetamorphic type, which was affected by significant polymetamorphism in the Caledonian and Variscan stages [15] according to the hypotheses made by [16,17].

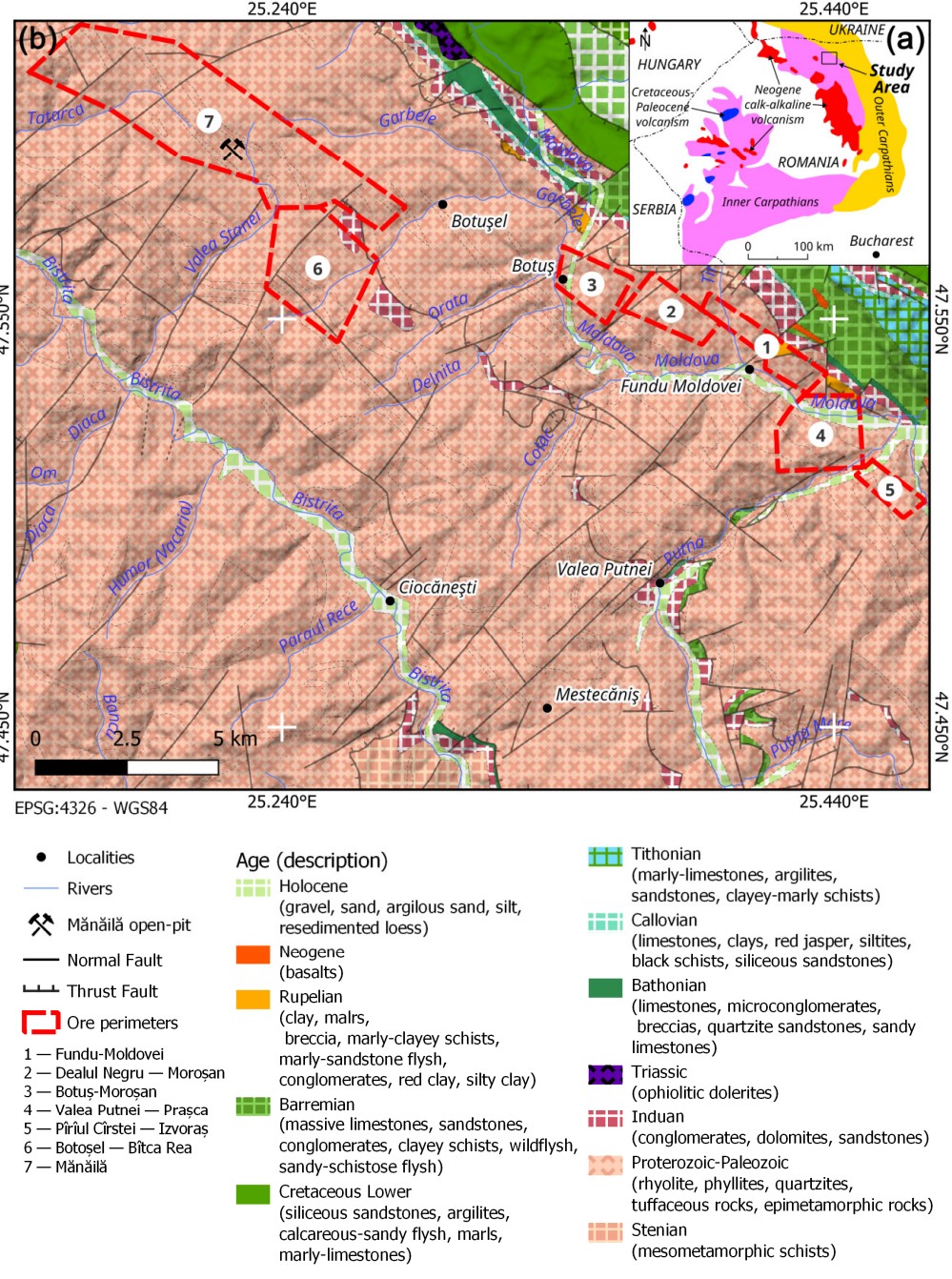

**Figure 1.** Localization of the studied area in the Carpathians, redrawn after Alderton and Fallick [18] (**a**); geological map of the Fundu Moldovei member with ore perimeters—modified after [19] (**b**).

The Tg1 formation is predominantly composed of quartzite sericite and quartzite sericite-chlorite schist over which quartzite and quarzto-feldspathic rocks are arranged.

Black quartzites, graphite schist, and limestone represent the Tg2 formation. The Tg3 formation has a wide development in the Fundu Moldovei and is lithostratigraphically constituted by four members: Isipoaia, Leșul Ursului, Moroșan, and Fundu Moldovei (Figure 2b). The last two members are predominant because the Isipoaia and Leșul Ursului members with great development south of Fundu Moldovei gradually disappear towards the NW, being cut by the Bucovinian overthrust. These formations are composed of rhyolitic metavolcanites separated by quartzo-feldspathic rocks and sericite-chlorite schists (Figure 1). Formation Tg4 consists of sericite-graphite schists and sericite-chlorite schists. Within this formation, several members were separated: Bașca, Pârâul Crucii, Afinet, Botuș, and Pângărați. The isotopic data presented by Kräutner [10] and Zincenco [15] attribute a lower Paleozoic age to the Tulgheș Lithogroup, an age also attested by Balintoni [4].

## 3. Materials and Methods

During the fieldwork conducted in 2018, 28 samples were collected from the four textural and mineralogical types. The optical microscopy observations were made in reflected light with the Meiji Techno model ML 9430 microscope. The equipment used for mineral identification were the SM-Plan of 4X, 10X, and 20X and the HWF10X eyepiece.

The chemical compositions were determined using a Cameca SX-100 Electron Probe Microanalyzer (EPMA). The analyses were carried out at the Earth Science Institute, Slovak Academy of Sciences (Banská Bystrica, Slovakia). The analysis points were selected using the backscattered electron (BSE) images. The measurements were performed on polished carbon-coated sections using an acceleration voltage of 25 kV and a 15–20 nA beam current, 5 μm beam diameter, 20 s integration time for the peak, and 7 s for the background. The following X-ray lines from the natural (n) and synthetic (s) standards were used: S $K\alpha$ (n-CuFeS$_2$), Zn $K\alpha$ (n-ZnS), Fe $K\alpha$ (n-CuFeS$_2$), Cu $K\alpha$ (n-CuFeS$_2$), Pb $M\alpha$ (n-PbS), As $K\alpha$, $L\alpha$ (n-FeAsS), Sb $L\beta$ (n-Sb$_2$S$_3$), Se $L\beta$ (s-Bi$_2$Se$_3$), and pure metals for Mn $K\alpha$, Cd $L\alpha$, Ag $L\alpha$, Bi $L\alpha$, and Te $L\alpha$.

The Raman spectra were obtained using the micro-Raman spectrometer Renishaw InVia coupled with a Peltier-cooled CCD detector, with a 532 nm laser with a nominal power of 50 mW at room temperature. The laser power was controlled by means of a series of density filters in order to avoid heating effects. The spectral interval was 67 to 1836 cm$^{-1}$ at a nominal spectral resolution of about 1.5 cm$^{-1}$. The micro-Raman spectra were obtained with an exposure time of 5 s, and 30 acquisitions at a laser power of 0.5%–1% in order to improve the signal-to-noise ratio. Data acquisition was carried out using Wire software.

The abbreviations of minerals reported in the present study follow the list approved by the International Mineralogical Association (IMA) Commission on New Minerals, Nomenclature and Classification (CNMNC) [20].

## 4. Results

### 4.1. Mănăilă Ore Deposit

The massive polymetallic pyrite mineralizations in the Fundu Moldovei field present a synclinal structure fragmented in some perimeters by the Bucovinian Nappe plane [6]. The mineralizations appear in three stratigraphic positions, separated by rhyolitic metatuffs and quartz-sericite schists (Figure 2a) [6,7]. The VMS mineralizations are stratiform which can be traced over distances of the order of meters (zone I and II) or in the form of stratum lenses (zone I). Several plots are known in the Fundu Moldovei field: Fundu Moldovei—Breaza, Dealul Negru-Moroșan, Botuș-Moroșan, Valea Putnei-Prașca, Cirstei-Izvoraș, and Mănăilă. There are three types of ore: compact, precompact, and impregnation (Figure 2c).

The Mănăilă ore deposit appears in the northern part of the Fundu Moldovei field and is confined in a lithological sequence attributed to the Moroșan, Fundu Moldovei, and Bacșa members of the Bucovinian Nappe. The structure is monoclinal with a NW-SE direction and slopes to the NE. The mineralization of polymetallic sulfides is located at two stratigraphic levels: an upper one (Fundu Moldovei sulfide horizon) and a lower one (Mănăilă type). The sulfide horizon of Fundu Moldovei has a modest mineralization represented mainly by

pyrite disseminations, rarely appearing at submetric levels with chalcopyrite and would be equivalent to the third level from Dealul Negru (Fundu Moldovei).

The upper level is cantoned in the upper part of the Fundu Moldovei sulfide horizon, which has rhyolitic metatuffs of the Fundu Moldovei in the footwall and sericite-graphite schist in the hanging wall. The mineralization in the form of strata lens is modest, represented by disseminations of pyrite with very small amounts of chalcopyrite.

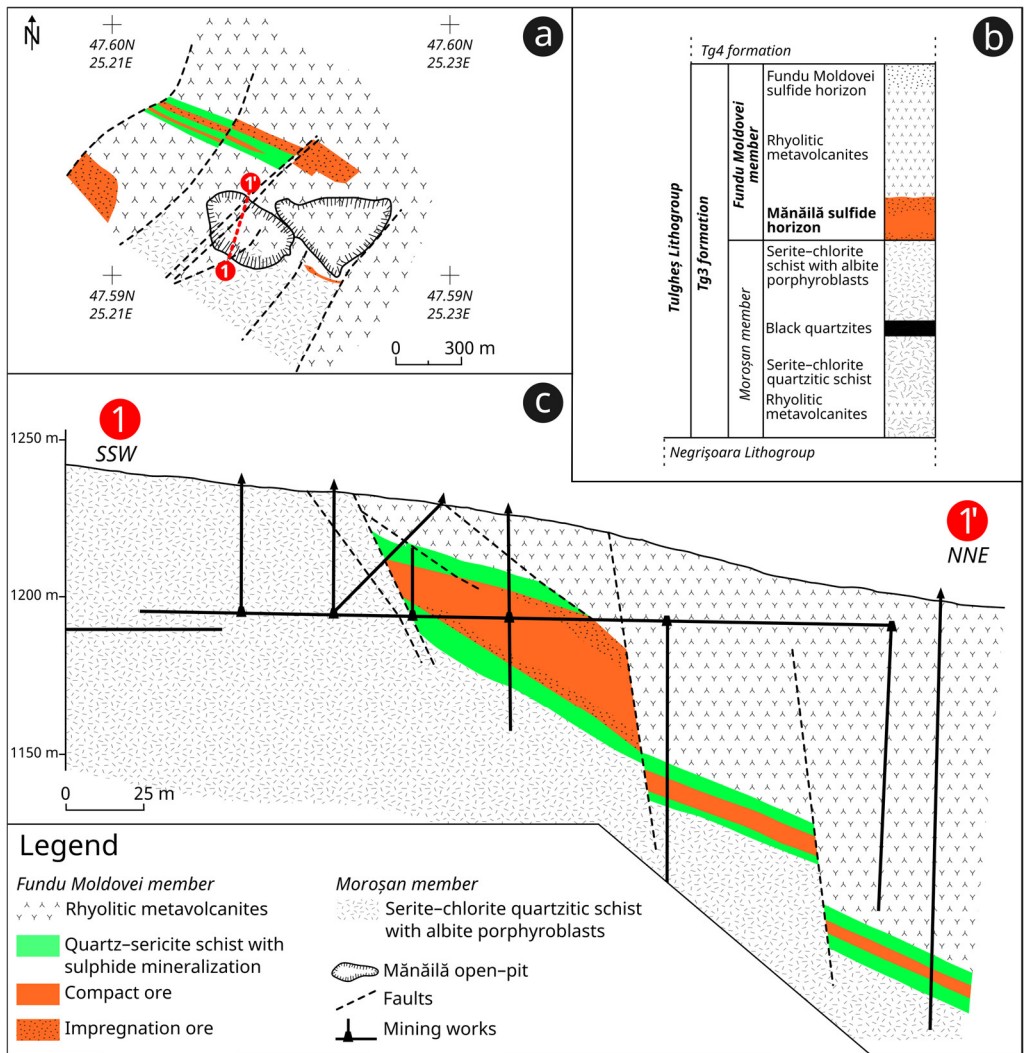

**Figure 2.** Simplified geological map of the Mănăilă ore deposit—modified after Harvey [21] (**a**); simplified lithostratigraphic sequence—modified after [10] (**b**); geological cross-section through the Mănăilă ore deposit (**c**).

The lower level (Mănăilă) is of particular importance, being located on the border between the Moroșan and Fundu Moldovei members. The ore is contained in a package of sericitic ± quartzitic shale (Mănăilă level) assigned to the Fundu Moldovei member. This package has rhyolitic metatuffs (which would belong to the Fundu Moldovei member) in its roof, on the basis of which, a landmark layer of chlorite-feldspathic metaepiclastics is identified, and in the footwall, a landmark level of chlorite quartzite schist and chlorite-sericite schist with albite porphyroblasts belonging member Moroșan. In the lower part of the Moroșan member, in the level of chlorite-quartzite schists, there are levels of black graphite-quartzites and white sericite-quartzites.

The chlorite-sericite and sericite-chlorite quartzite schists in the Moroșan member are mainly composed of quartz, chlorite, and sericite with obvious schistosity and lepidogranoblastic structure, highlighted by an alternation of millimetric levels of quartz and sericite.

At the upper part of the level, albite microporphyroblasts can be observed with the naked eye or with a magnifying glass, passing to sericite-chlorite quartzite schists with albite porphyroblasts. The rhyolitic metatuffs that cover the sulfide level are quartz-feldspathic rocks with sericite showing a less obvious schistosity. The quartz and albite form blasto-porphyres visible to the naked eye. The rocks that surround the mineralized level from Mănăilă were metamorphosed into the greenschists facies, the chlorite zone.

The form of the deposit is an elongated lentil concordant with the epimetamorphic schists folded synchronously with the rocks from the footwall and hanging wall. The strata lens is in the form of a monocline with a general NW-SE direction and northeast inclinations of 20–40°, compartmented by the direction and inclination of the faults. The transverse faults have a north-west inclination and produce directional compartments with variable extensions and vertical displacements reaching tens of meters. The thickness of the mineralization is between 1 and 19 m (Figures 2c and 3).

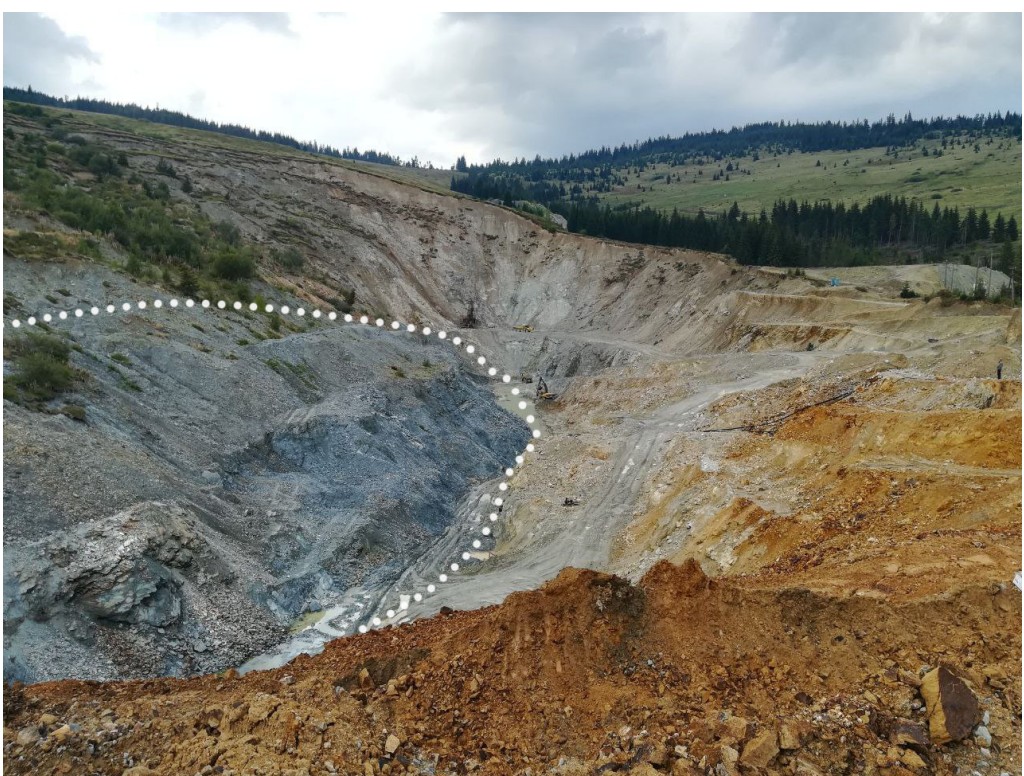

**Figure 3.** Panoramic view of the Mănăilă open pit (photo date: September 2018). The dotted line represents the mineralization presented in rhyolitic metatuffs.

The mineralogical composition common to all three ore types is relatively simple: pyrite, chalcopyrite, sphalerite, and galena. Small amounts of arsenopyrite, tetrahedrite, bornite, and covellite were pointed out by [22] in paragenesis with quartz, chlorite, and muscovite. The gangue minerals consist of quartz, mica, carbonates, and barite. During our research, we also identified other minerals such as wittichenite, mawsonite, and an unnamed mineral (see further discussions).

The ore deposit is not very large and contains Cu = 2.23%, Pb = 0.82%, Zn = 1.86%, and S = 28.76%. The deposit was mined intermittently in an open pit between 1988 and 2019.

### 4.2. Textural and Mineralogical Types of Ores

The texture and structure of the ore show the imprint of the metamorphism. There are three textural types of ore: compact, precompact, and disseminated with graded transitions from one type to another. The disseminated type of ore is located outside the compact one. The marginal zone of the strata lens and disseminated ore were both found to have a

schistose texture. The following types of ore were identified based on optical observations: pyrite-polymetallic, pyrite copper, compact and precompact copper, and quartz-precompact copper. The mineralizations in the Mănăilă area are characteristic of volcanogenic massive sulfide (VMS) deposits in terms of host formations, textures, and chemistry.

The polymetallic pyrite ore is mainly composed of compact pyrite with galena, sphalerite, and chalcopyrite (Figure 4a). Microscopically, arsenopyrite, tetrahedrite, chalcocite, and covellite were identified. The texture is compact with a submillimeter granulation with subhedral and anhedral pyrite crystals and with anhedral grains of sphalerite, chalcopyrite, and galena arranged in the free spaces between the pyrite grains.

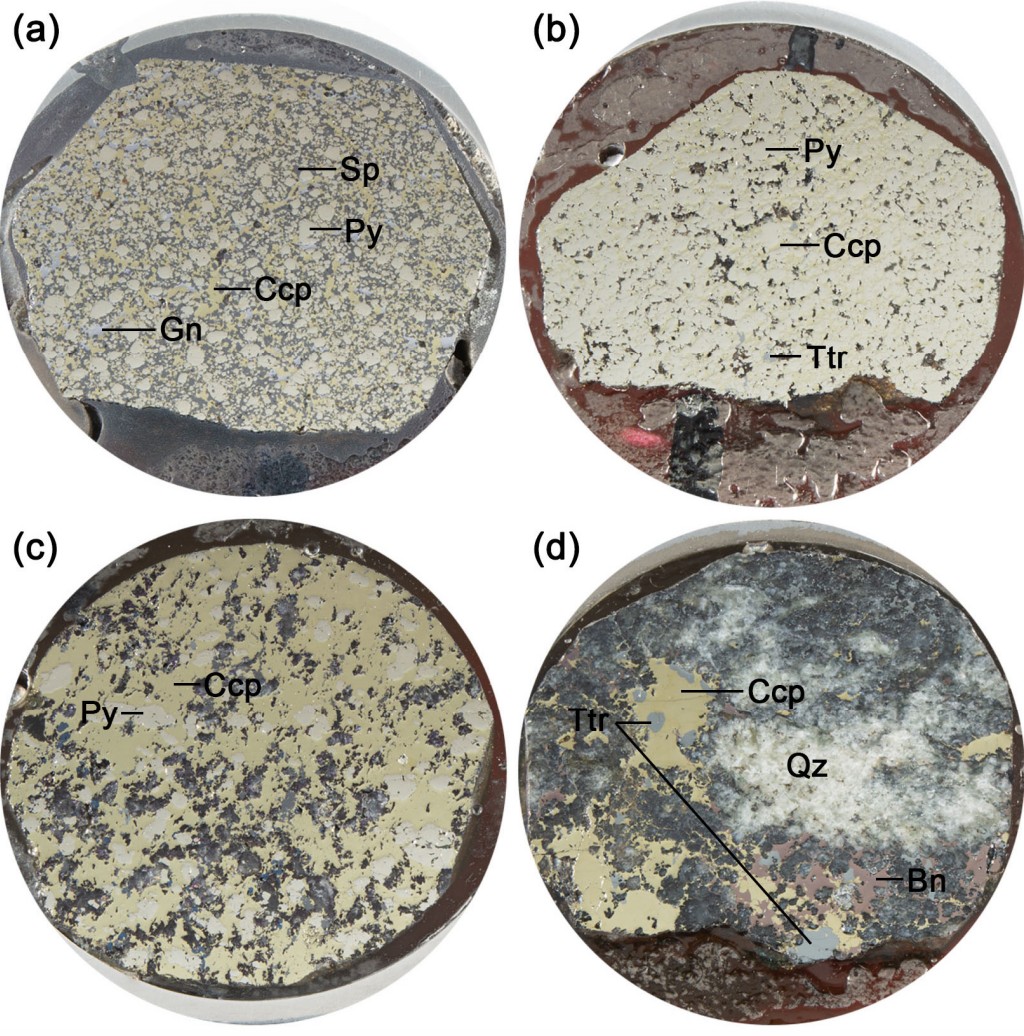

**Figure 4.** Macrophotographs of the representative mineral assemblages of the mineralizations in the Fundu Moldovei: pyrite-polymetallic type with subhedral pyrite grains cemented by sphalerite, chalcopyrite and galena (**a**); pyrite copper type with subhedral pyrite grains cemented by chalcopyrite and tetrahedrite (**b**); compact copper type with compact chalcopyrite with anhedral inclusions of pyrite and quartz (**c**); and precompact quartz-copper type with quartz that includes anhedral grains of chalcopyrite, tetrahedrite, and bornite (**d**). Sp—sphalerite; Py—pyrite; Ccp—chalcopyrite; Gn—galena; Ttr—tetrahedrite; Qz—quartz; Bn—bornite. Scale bar—1 cm.

The copper pyrite ore (Figure 4b) is mostly formed of pyrite in the form of submillimeter subhedral and anhedral grains, with chalcopyrite molded through the open spaces, while the chlorite and sericite grains appear between the pyrite grains. Tetrahedrite is frequently associated with chalcopyrite. This type of ore is quite common in precompact ore.

The compact and precompact copper ore consists predominantly of compact masses of chalcopyrite associated with subordinate pyrite. Sphalerite occurs in the form of submillimeter grains included in pyrite. Covellite replaces chalcopyrite and appears as thin layers deposited on the sphalerite. Quartz, chlorites, and sericites appear as nests included in the chalcopyrite.

The precompact quartz-copper ore consists of chalcopyrite, bornite, tetrahedrite, chalcocite, covellite, pyrite, and galena. The pyrite is frequently brecciated and cemented by subsequent deposits of bornite, chalcopyrite, and tetrahedrite. This textural type suggests the formation of this ore in epigenetic conditions as a result of copper remobilization by metamorphic solutions. The metallic minerals are included in compact masses of quartz, sericite, and small amounts of barite. This type of ore is not found in other deposits from the massive pyrite belt in the Eastern Carpathians previously described by [7,9].

### 4.3. Ore Mineralogy of Mănăilă Level

#### 4.3.1. Sulfides

Pyrite, chalcopyrite, sphalerite, and galena are major minerals; bornite, covellite, tetrahedrite, chalcocite, and arsenopyrite are minor, with accessory minerals of bismuth and tin.

Pyrite appears in a compact form in pyrite-polymetallic (Figure 5a) and pyrite-copper ore (Figure 5b,d) and is the common ore mineral in precompact and disseminated ore. Pyrite can be found in the form of subhedral and anhedral equigranular grains (Figure 5a–c). Pyrite is spatially associated with copper-bearing sulfides, sphalerite, and galena (Figure 5a–c). Pyrite grains are corroded by chalcopyrite and sometimes by sphalerite. In massive ores, fine-grained pyrite aggregates are predominant. Pyrite also has brecciated structures in the spaces created, more plastic chalcopyrite infiltrates under the influence of stress (Figure 5a,b). Pyrite in compact ore is spatially associated with copper-bearing sulfides and sphalerite-galena.

Electron microprobe analysis indicates minor amounts of As, Ni, Co, Pb, Ag, and Mn in the pyrite (Table 1). The pyrite is poor in As (<0.06 wt.%) but contains up to 0.05 wt.% Co; the other elements are insignificant. Impurities in the pyrite are incorporated into the crystal structure.

**Table 1.** Representative EPMA results (wt.%) of common sulfides from the Mănăilă ore deposit: pyrite, sphalerite, and galena.

| Sample Name | S | Fe | Pb | Zn | Ag | Cu | Mn | As | Co | Ni | Cd | Mn | Bi | Sb | Total | Mineral |
|---|---|---|---|---|---|---|---|---|---|---|---|---|---|---|---|---|
| MN-5_an3 | 54.39 | 47.24 | 0.29 | n.d. | 0.04 | 0.01 | 0.03 | 0.05 | 0.04 | n.d. | n.d. | n.d. | n.d. | n.d. | 102.09 | Pyrite |
| MN-8_an10 | 54.06 | 46.93 | 0.28 | n.d. | 0.03 | 0.02 | n.d. | 0.06 | 0.05 | 0.02 | n.d. | n.d. | n.d. | n.d. | 101.44 | Pyrite |
| MN-8_an17 | 53.88 | 46.61 | 0.26 | n.d. | 0.06 | 0.02 | n.d. | 0.06 | 0.05 | n.d. | n.d. | n.d. | n.d. | n.d. | 100.94 | Pyrite |
| MN-5_an2 | 33.88 | 0.22 | 0.13 | 67.67 | 0.02 | 0.05 | n.d. | n.d. | n.d. | n.d. | 0.32 | 0.03 | n.d. | n.d. | 102.33 | Sphalerite |
| MN-5_an4 | 14.02 | n.d. | 88.54 | n.d. | 0.08 | 0.03 | n.d. | n.d. | n.d. | n.d. | n.d. | n.d. | 0.38 | 0.01 | 103.05 | Galena |
| MN-5_an5 | 14.00 | n.d. | 88.94 | n.d. | 0.03 | 0.04 | n.d. | n.d. | n.d. | n.d. | n.d. | n.d. | 0.32 | 0.05 | 103.37 | Galena |
| Mn-8 an20 | 13.15 | n.d. | 85.45 | n.d. | 0.10 | n.d. | n.d. | n.d. | n.d. | n.d. | n.d. | n.d. | 0.35 | n.d. | 99.05 | Galena |

n.d. = not detected.

Sphalerite is more abundant in pyrite-polymetallic ore (Figure 4a) than in copper ore (Figure 4c). In the pyrite-polymetallic ore, it appears in the form of anhedral crystals of up to 1 mm intergrowth with chalcopyrite and galena and is arranged in the open spaces between the pyrite grains (Figures 4a and 5c). Electron microprobe analysis (Table 1) indicates sphalerite with low Fe and Mn content and 0.32 wt.% Cd content. Sphalerite contains impurities of Cu, Pb, and Ag, but no inclusions of other minerals were observed in the analyzed area.

Galena is a minor constituent intergrowth with sphalerite and chalcopyrite in pyrite-polymetallic ore (Figures 4a and 5a) and inclusions in precompact quartz-copper ore. The silver content reaches up to 0.1 wt.% and the bismuth content up to 0.38 wt.% (Table 1). The presence of copper was also highlighted but in insignificant quantities. These impurities are most likely present in the crystalline structure since no other minerals were observed as inclusions in the microprobe images.

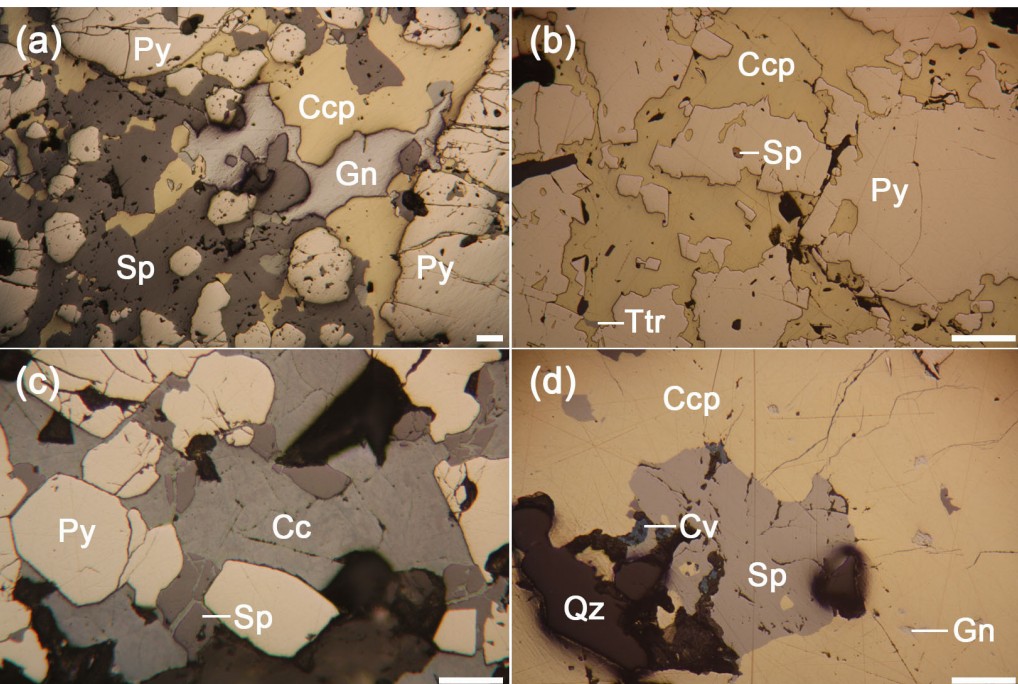

**Figure 5.** Plane-polarized reflected light photomicrographs of representative mineral assemblages from Mănăilă ore deposit: pyrite-polymetallic type with euhedral and subhedral grains of pyrite cemented and intergrowth with chalcopyrite, sphalerite, and galena; the cracks in the pyrite are filled with chalcopyrite (**a**); pyrite copper type, anhedral pyrite corroded and intergrowth with chalcopyrite; frequent inclusions of chalcopyrite and sphalerite in pyrite; and micrometric chalcopyrite veins in pyrite (**b**); pyrite-polymetallic type with euhedral and anhedral pyrite cemented by sphalerite and chalcocite; the sphalerite is brecciated and cemented by chalcocite (**c**); pyrite copper type, compact masses of chalcopyrite with inclusions of sphalerite, galena and quartz; the sphalerite is replaced by secondary covellite (**d**). Sp—sphalerite; Py—pyrite; Ccp—chalcopyrite; Gn—galena; Ttr—tetrahedrite; Qz—quartz; Cc—chalcocite; Cv—covellite. Scale bars on the photomicrographs are 100 μm.

Chalcopyrite is abundant in the Mănăilă ore deposit and occurs in high amounts in pyrite-copper (Figure 4b), compact and precompact copper (Figure 4c), and quartz-copper ore (Figure 4d). It forms anhedral grains that border pyrite or compact masses with sphalerite, pyrite, galena, and tetrahedrite inclusions (Figures 5d and 6a–d). Chalcopyrite is frequently intergrown with bornite and tetrahedrite in quartz-copper precompact ore (Figure 6e,f). Chalcopyrite crosscuts the pyrite grains from compact pyrite ore (Figure 5a). Electron microprobe data indicate minor amounts of As, Sn, Co, Ni, Mn, Bi, Pb, Zn, and Cd in the crystal structure of chalcopyrite (Table 2).

Bornite is most abundant in the precompact quartz-copper ore and is present as anhedral millimetric grains intergrown with chalcopyrite (Figure 6e) or as small inclusions in chalcopyrite. Very frequently, bornite crosscuts pyrite grains (Figure 7a) or includes tetrahedrite grains (Figure 7b) and chalcocite (Figure 7c,d) in precompact quartz-copper ore. Bornite is replaced and corroded by covellite (Figure 7d). Bornite contains up to 0.61 wt.% Ag and very minor amounts of Co, As, Sn, Ni, Bi, Sb, Te, Pb, Zn, and Cd (Table 2). The microscopic textures (Figure 7a) indicate the subsequent deposition of bornite after the regional metamorphism process.

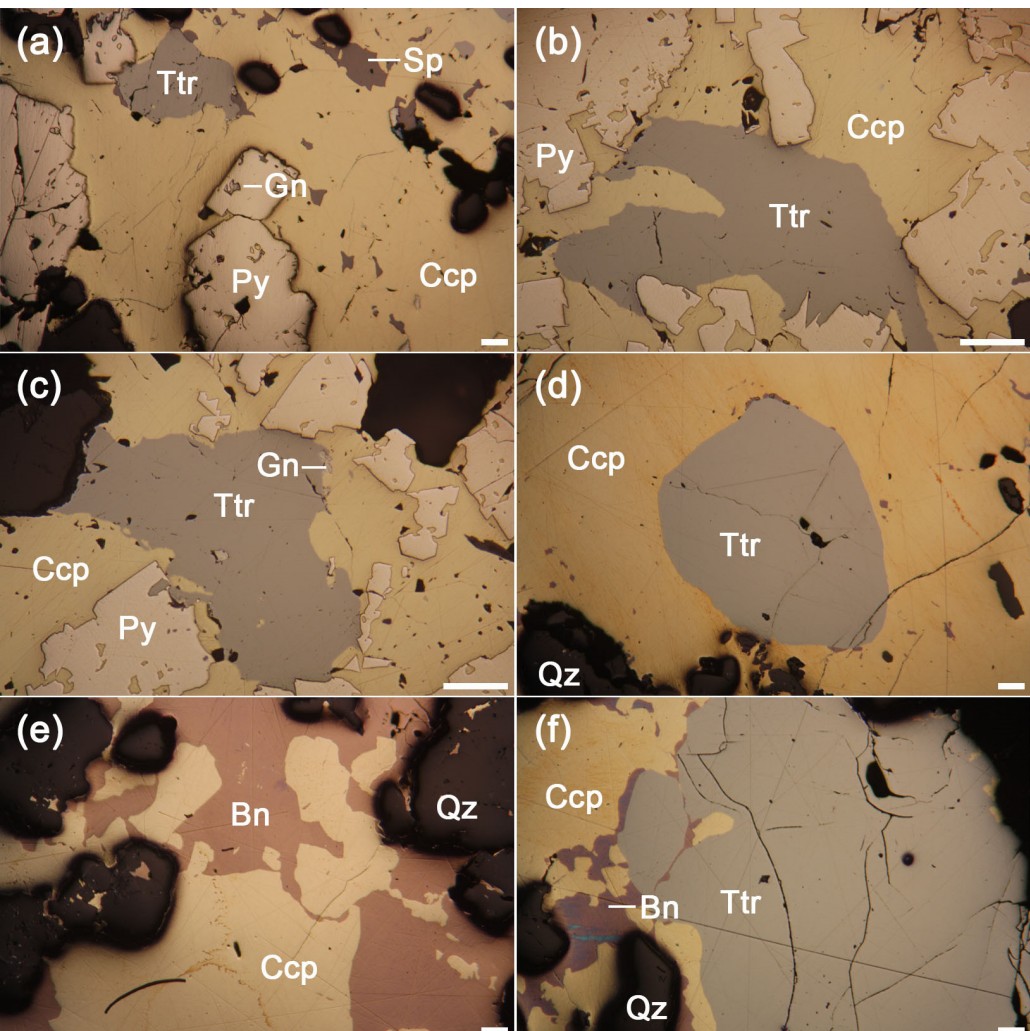

**Figure 6.** Plane-polarized reflected light photomicrographs of representative copper mineral assemblages: pyrite copper with anhedral and euhedral pyrite cemented and corroded by chalcopyrite; chalcopyrite filling of pyrite pore space (**a**–**c**); anhedral grains of tetrahedrite included in chalcopyrite (**b**–**d**); quartz-precompact copper with an intergrowth chalcopyrite, bornite, and tetrahedrite, arranged in the free spaces between the quartz (**e**,**f**). Sp—sphalerite; Py—pyrite; Ccp—chalcopyrite; Gn—galena; Ttr—tetrahedrite; Qz—quartz; Bn—bornite. Scale bars on the photomicrographs are 100 μm.

Primary chalcocite is common in precompact quartz-copper ore along with chalcopyrite and bornite. It appears as an inclusion in the bornite and is corroded marginally and on cracks by covellite (Figure 7c,d). Covellite rims are common in contact between bornite and chalcocite (Figure 7d). Chalcocite and bornite frequently intergrow each other (Figure 8a,b). Secondary chalcocite formed by the weathering of chalcopyrite (Figure 8c) and as a pellicular form of sphalerite grains (Figure 5c) has been identified in pyrite-polymetallic ore. The chemical composition of the two types of chalcocite is not much different, however, the difference can be noticed in the silver content; primary chalcocite contains more silver (0.76–1.89 wt.%) than secondary (0.35–0.49 wt.%) (Table 2). The minor elements are also different, in secondary chalcocite Co is frequently present, and primary chalcocite contains As, Pb, and Zn.

**Table 2.** Representative EPMA results (wt.%) of copper sulfides from the Mănăilă ore deposit: chalcopyrite, bornite, chalcocite, and covellite.

| Sample Name | S | Cu | Fe | Ag | As | Sn | Co | Ni | Mn | Bi | Sb | Te | Pb | Zn | Cd | Total | Mineral |
|---|---|---|---|---|---|---|---|---|---|---|---|---|---|---|---|---|---|
| MN-9_an2 | 34.61 | 34.94 | 30.62 | 0.04 | 0.05 | n.d. | n.d. | 0.01 | n.d. | 0.14 | n.d. | n.d. | 0.18 | 0.04 | 0.01 | 100.646 | Chalcopyrite |
| MN-14_an2 | 35.34 | 34.54 | 30.84 | 0.03 | 0.03 | n.d. | 0.01 | n.d. | n.d. | 0.16 | n.d. | 0.03 | 0.15 | 0.05 | 0.01 | 101.197 | Chalcopyrite |
| MN-14_an4 | 35.05 | 34.69 | 30.84 | 0.03 | n.d. | n.d. | 0.02 | n.d. | n.d. | n.d. | n.d. | n.d. | n.d. | n.d. | n.d. | 100.629 | Chalcopyrite |
| MN-8_an2 | 25.94 | 62.99 | 11.39 | 0.61 | 0.02 | n.d. | n.d. | n.d. | n.d. | n.d. | n.d. | n.d. | n.d. | n.d. | n.d. | 100.954 | Bornite |
| MN-8_an8 | 25.53 | 63.78 | 10.82 | 0.44 | n.d. | n.d. | n.d. | n.d. | n.d. | n.d. | n.d. | n.d. | n.d. | n.d. | n.d. | 100.57 | Bornite |
| MN-8_an9 | 25.91 | 63.27 | 11.5 | 0.33 | n.d. | n.d. | n.d. | 0.02 | n.d. | n.d. | n.d. | n.d. | n.d. | n.d. | n.d. | 101.03 | Bornite |
| MN-8_an11 | 25.94 | 62.99 | 11.32 | 0.36 | n.d. | n.d. | 0.01 | n.d. | n.d. | 0.02 | 0.05 | 0.01 | 0.14 | 0.18 | 0.01 | 101.028 | Bornite |
| MN-8_an15 | 26.34 | 63.67 | 11.59 | 0.31 | n.d. | n.d. | n.d. | n.d. | n.d. | n.d. | n.d. | n.d. | n.d. | n.d. | n.d. | 101.912 | Bornite |
| MN-8_an16 | 26.08 | 63.06 | 11.46 | 0.31 | n.d. | n.d. | 0.01 | n.d. | n.d. | n.d. | n.d. | n.d. | n.d. | n.d. | n.d. | 100.916 | Bornite |
| MN-14_an3 | 26.18 | 62.81 | 11.5 | 0.3 | n.d. | n.d. | n.d. | n.d. | n.d. | n.d. | n.d. | n.d. | n.d. | n.d. | n.d. | 100.79 | Bornite |
| MN-14_an5 | 26.38 | 62.82 | 12.06 | 0.37 | n.d. | n.d. | 0.03 | n.d. | n.d. | n.d. | n.d. | n.d. | n.d. | n.d. | n.d. | 101.656 | Bornite |
| MN-7_an1 | 21.86 | 74.01 | 0.26 | 0.5 | n.d. | n.d. | n.d. | n.d. | n.d. | n.d. | 0.01 | n.d. | n.d. | n.d. | n.d. | 96.643 | Chalcocite |
| MN-7_an2 | 21.68 | 75.35 | 0.42 | 0.35 | n.d. | n.d. | 0 | n.d. | n.d. | n.d. | n.d. | n.d. | n.d. | n.d. | n.d. | 97.802 | Chalcocite |
| MN-7_an3 | 21.02 | 76.58 | 1.03 | 0.39 | n.d. | n.d. | 0.01 | n.d. | n.d. | n.d. | n.d. | n.d. | n.d. | n.d. | n.d. | 99.026 | Chalcocite |
| MN-7_an4 | 21.69 | 75.87 | 0.48 | 0.44 | n.d. | n.d. | 0 | n.d. | n.d. | n.d. | n.d. | n.d. | n.d. | n.d. | n.d. | 98.48 | Chalcocite |
| MN-7_an5 | 21.08 | 76.44 | 0.64 | 0.49 | n.d. | n.d. | 0.01 | n.d. | n.d. | n.d. | n.d. | n.d. | n.d. | n.d. | n.d. | 98.656 | Chalcocite |
| MN-8_an3 | 21.34 | 78.28 | 0.36 | 0.84 | 0.01 | n.d. | n.d. | n.d. | n.d. | 0.05 | n.d. | 0.01 | 0.07 | 0.08 | n.d. | 101.038 | Chalcocite |
| MN-8_an7 | 21.14 | 78.96 | 0.16 | 0.76 | 0.01 | n.d. | n.d. | n.d. | n.d. | n.d. | n.d. | n.d. | 0.08 | 0.04 | n.d. | 101.152 | Chalcocite |
| MN-8_an13 | 21.66 | 75.79 | 2.89 | 1.89 | 0.03 | n.d. | n.d. | n.d. | n.d. | n.d. | n.d. | n.d. | 0.09 | 0.05 | n.d. | 102.397 | Chalcocite |
| MN-8_an14 | 22.38 | 75.51 | 2.992 | 1 | n.d. | n.d. | n.d. | 0.01 | n.d. | n.d. | n.d. | n.d. | n.d. | n.d. | n.d. | 101.889 | Chalcocite |
| MN-1_an3 | 34.33 | 67.12 | 0.19 | n.d. | n.d. | n.d. | n.d. | 0.01 | n.d. | n.d. | n.d. | n.d. | n.d. | n.d. | n.d. | 101.646 | Covellite |

n.d. = not detected.

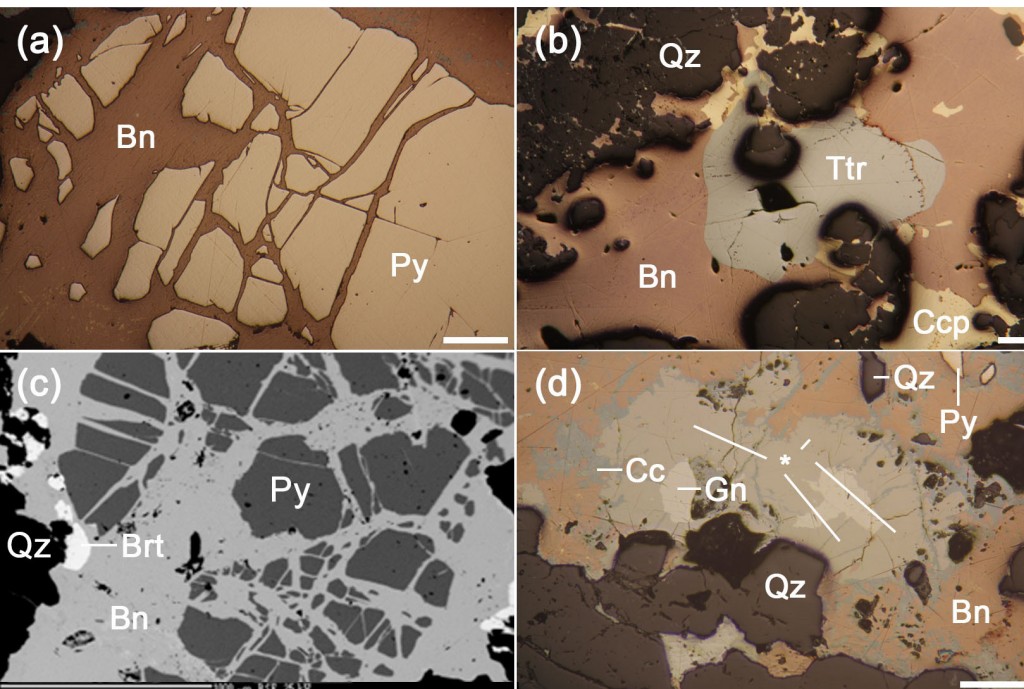

**Figure 7.** Plane-polarized reflected light photomicrographs and back-scattered electron images showing representative copper mineral assemblages from quartz-precompact copper: pyrite brecciated and cemented by bornite (**a**); bornite intergrowth with quartz and tetrahedrite inclusions (**b**); pyrite brecciated and cemented by bornite; barite at the border of bornite and quartz (**c**); intergrowth between bornite and the unnamed mineral; secondary replacement of chalcocite on cracks and at the border of unnamed mineral; inclusions of galena in the unnamed mineral (**d**). Bn—bornite; Py—pyrite; Ttr—tetrahedrite; Qz—quartz; Gn—galena; Cc—chalcocite; *—unnamed mineral, Brt -barite. Scale bars on the photomicrographs are 100 μm.

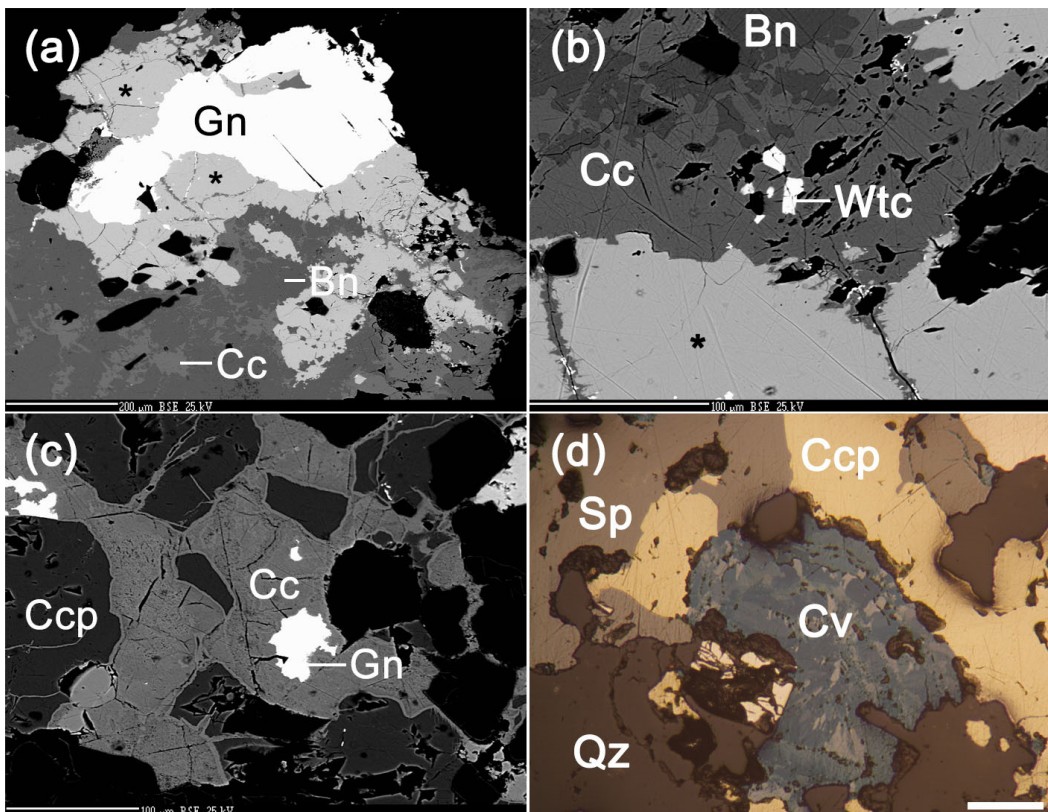

**Figure 8.** Plane-polarized reflected light photomicrographs and back-scattered electron images showing the unnamed mineral and copper minerals from quartz-precompact copper: intergrowth between the unnamed mineral, chalcocite, bornite, and galena (**a**); intergrowth between the unnamed mineral, chalcocite, and bornite; wittichenite is present as small inclusions in the chalcocite (**b**); secondary chalcocite with galena inclusions that replace and cement chalcopyrite (**c**); secondary covellite substitutes the intergrowth between chalcopyrite and sphalerite (**d**). Gn—galena; *—unnamed mineral; Cc—chalcocite; Bn—bornite; Wtc—wittichenite; Ccp—chalcopyrite; Sp—sphalerite; Cv—covellite; Qz—quartz. Scale bars on the photomicrographs are 100 μm.

Covellite is an accessory mineral formed by the alteration of primary copper sulfides. It appears as thin rims at the bornite–chalcocite contact (Figure 7d), or on sphalerite and chalcopyrite (Figure 8d). The chemical composition is simple (CuS), with few impurities (Fe, Ni) (Table 2).

The unnamed mineral from Mănăilă was identified in a less typical association with bornite, chalcocite, and galena (Figures 7d and 8a,b). Chalcocite and the unnamed mineral are substituted on cracks by covellite. According to this association, the unnamed mineral is a primary mineral and not a secondary one. The relief of the unnamed mineral is close to chalcocite and bornite and higher than galena. Its color is cream (Figure 7d), with weak bireflection, and strong anisotropy with colors from gray to brown. The reflectivity is lower than galena and higher than bornite and chalcocite.

The electronic microprobe data of the unnamed mineral (Table 3) indicate the presence of large amounts of Cu, Pb, and S. In smaller amounts, Fe and Ag substitute for Cu and Pb, respectively. Minor amounts of Zn, As, Cd, and Bi were also observed. The calculated formula for the unnamed mineral would be the following: $Cu_{20.91}Fe_{0.99}Zn_{0.01}Ag_{0.22}Pb_{1.86}S_{13.99}$.

**Table 3.** Electron microprobe analyses for the unnamed mineral (wt.%).

| Sample Name | S | Fe | Cu | Zn | As | Ag | Cd | Te | Pb | Bi | Total |
|---|---|---|---|---|---|---|---|---|---|---|---|
| MN-8_an1 | 20.68 | 2.46 | 59.23 | 0.08 | 0.04 | 1.01 | n.d. | n.d. | 17.55 | n.d. | 101.05 |
| MN-8_an6 | 21.02 | 2.49 | 60.04 | 0.03 | n.d. | 0.80 | 0.01 | 0.02 | 17.75 | 0.07 | 102.23 |
| MN-8_an18 | 19.28 | 2.49 | 59.78 | 0.04 | n.d. | 0.93 | n.d. | 0.01 | 16.85 | 0.03 | 99.41 |
| MN-8_an19 | 19.54 | 2.47 | 58.81 | 0.04 | n.d. | 1.52 | 0.04 | n.d. | 16.79 | 0.13 | 99.34 |
| MN-8_an21 | 19.84 | 2.41 | 59.33 | 0.02 | 0.01 | 1.14 | 0.01 | n.d. | 17.44 | 0.03 | 100.23 |
| **Average** | **20.07** | **2.46** | **59.44** | **0.04** | **0.01** | **1.08** | **0.01** | **0.01** | **17.28** | **0.05** | **100.45** |

n.d. = not detected.

The Raman spectra of the unnamed mineral show the most intense peak at 206 cm$^{-1}$ (Figure 9a). No other bands were observed at higher wavenumbers, only in the low-frequency region where several lines appear at 73, 159–161, 173, and 194 cm$^{-1}$. Additional peaks are present as shoulders or with lower intensities (Figure 9a).

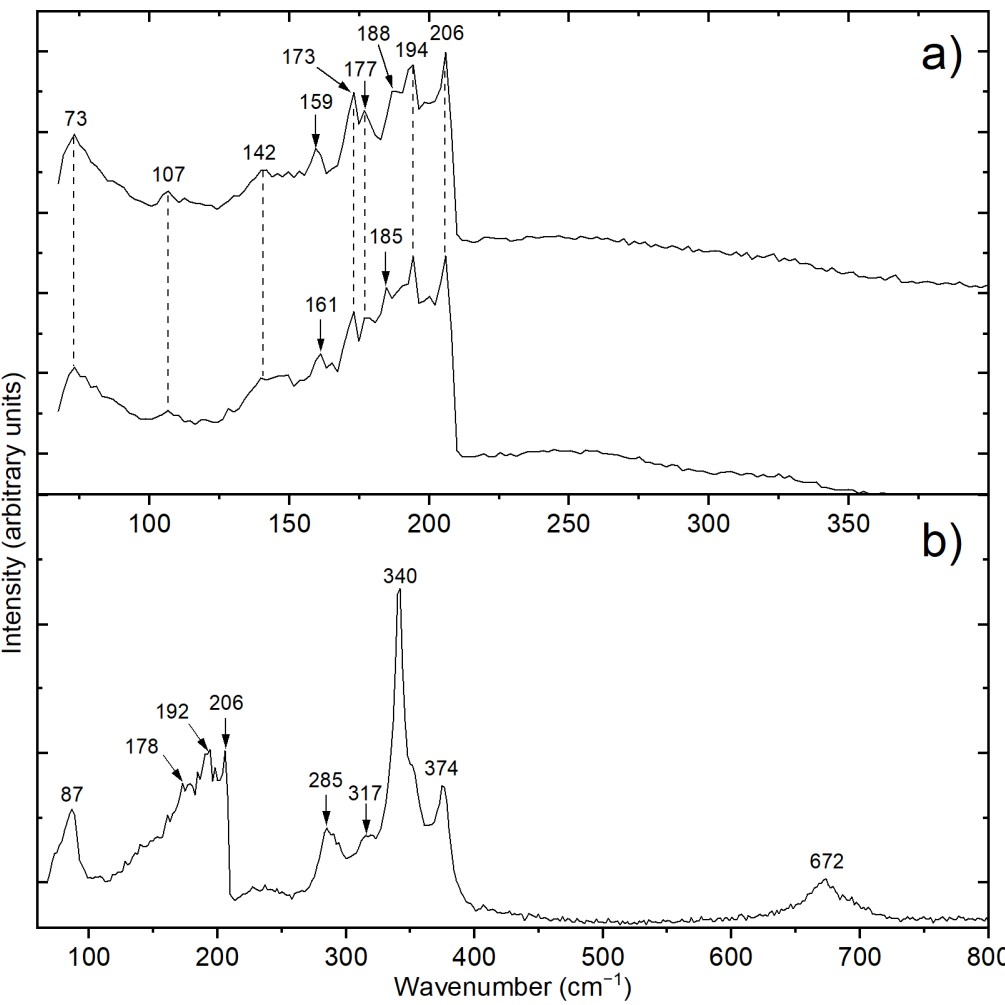

**Figure 9.** Selective Raman spectra of the unnamed mineral (**a**); the Raman spectrum of mawsonite (**b**).

### 4.3.2. Sulfosalts

Mawsonite appears in the form of submillimeter inclusions in bornite (Figure 10a,b). It is in the same association with many bornites as in the one described by Markham and Lawrence [23]. The color is orange-pink, much lighter than bornite, and with a lower relief (Figure 10a). It presents obvious bireflection and strong anisotropy with colors varying from light yellow to blue. The chemical analysis of mawsonite is presented in Table 4. The

crystallochemical formula is: $Cu_{6.01}Sn_1Fe_{2.03}Ag_{0.01}Pb_{0.02}Zn_{0.01}As_{0.01}S_{7.91}$ which is identical to the structural formula of mawsonite $Cu_6SnFe_2S_8$.

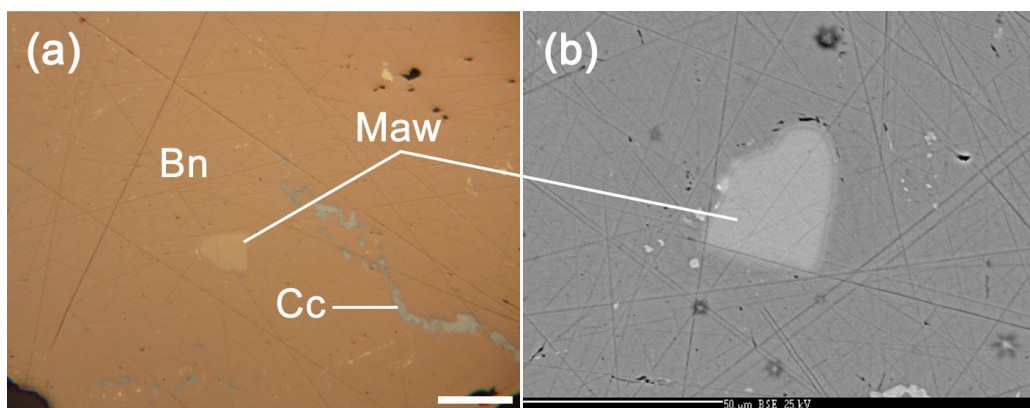

**Figure 10.** Plane-polarized reflected light photomicrographs (**a**) and back-scattered electron images of mawsomite (Maw) included in bornite (Bn) (**b**). Chalcocite (Cc) is also present. The scale bar on the photomicrograph is 100 µm.

**Table 4.** Electron microprobe analysis for mawsonite (wt.%).

| SampleN | Te | Cu | Sn | Fe | S | Ag | Pb | Zn | As | Cd | Total |
|---|---|---|---|---|---|---|---|---|---|---|---|
| MN-8 | 0.04 | 44.86 | 13.97 | 13.30 | 29.78 | 0.08 | 0.15 | 0.04 | 0.05 | 0.02 | 102.28 |

The Raman spectrum of mawsonite is shown in Figure 9b. The peaks are divided into four groups with the most intense one at 340 cm$^{-1}$ and the additional lines 285, 317, and 374 cm$^{-1}$. One vibrational mode appears at 672 cm$^{-1}$ while at lower wavenumbers, the Raman bands 178, 192, and 206 cm$^{-1}$ are observed. The lattice modes are shown at 87 cm$^{-1}$.

Wittichenite represents the first occurrence in the polymetallic massive sulfide mineralizations of the Eastern Carpathians, Romania. It appears in the form of 10–15 µm granules included in chalcocite (Figure 8b). Electron microprobe data (Table 5) indicate a wittichenite with small impurities of Sn, Fe, Ag, Sb, Pb, and Te that replace the main elements Cu, Bi, and S in its crystal structure. The chemical formula is $Cu_{3.11}Bi_{0.95}Ag_{0.01}Fe_{0.01}S_{2.94}$ which corresponds to the structural one of wittichenite.

**Table 5.** Electron microprobe analyses for wittichenite (wt.%).

| Analyses No. | Cu | Bi | S | Sn | Fe | Ag | Sb | Te | Pb | Total |
|---|---|---|---|---|---|---|---|---|---|---|
| 1 | 40.60 | 39.60 | 19.98 | 0.03 | 0.05 | 0.39 | n.d. | 0.01 | 0.11 | 100.77 |
| 2 | 42.98 | 38.59 | 19.89 | 0.01 | 0.12 | 0.59 | n.d. | n.d. | 0.10 | 102.28 |

Tetrahedrite is the most common mineral from the sulfosalt group present in the VMS deposits of the Eastern Carpathian belt [7,9,22,24]. For the natural minerals of the tetrahedrite-tennantite series ($Cu_{10}(Fe, Zn)_2Sb_4S_{13}$–$Cu_{10}(Fe, Zn)_2As_4S_{13}$) the term "fahlore(s)" is frequently used [25]. In the massive pyrite deposit in the Mănăilă perimeter, tetrahedrite appears in all types of mineralization, but especially in copper ores (Figure 4b–d). The specific presentation of tetrahedrite in the type of compact ore is in the form of granules included in chalcopyrite that constitutes the matrix of subhedral pyrite crystals (Figure 5a). In the copper ore, the tetrahedrite appears in the form of anhedral grains included in chalcopyrite (Figures 4b and 6b–d). More widely developed tetrahedrite grains of 2 mm are associated with or included in chalcopyrite and bornite (Figures 5d, 6f and 7b).

For tetrahedrites, 26 electron microprobe (EPMA) analyses were performed (Table 6). The tetrahedrite from Mănăilă does not show a great compositional range, being represented by the arsenic end-member (tennantite). Fe and Zn substitute for $Cu^{2+}$ in the structure of tennantite from Mănăilă and are negatively correlated ($r = 0.82$, Figure 11a). When the Fe *apfu* has values lower than 1, and if $Cu^{2+}$ is in excess, then Fe appears as $Fe^{3+}$ according to the data of [26]. Fe+Zn exceeds the 2 *apfu* value because it probably represents $Cu^{2+}$ in excess or $Cu^+$. Trace amounts of Mn, Pb, and Cd substitute for Fe and Zn in the divalent *C* sites, while low contents of Ag substitute replace $Cu^+$ up to 0.15 *apfu* in the tennantite structure. The predominant semimetal is As which appears between 3.10—4.15 *afpu*, indicating the presence of tennantite members. As expected, there is a strong negative correlation between As and Sb ($r = 0.88$, Figure 11b).

**Table 6.** Electron microprobe analyses of tennantites (wt.%).

| Sample Name | Te | Sb | Ag | Fe | Se | As | Cu | Mn | Bi | Cd | Pb | S | Zn | Sn | Hg | Total | Sb/(Sb + As) |
|---|---|---|---|---|---|---|---|---|---|---|---|---|---|---|---|---|---|
| MN-3 an 1 | n.d. | 2.39 | 0.06 | 1.86 | n.d. | 17.93 | 42.82 | 0.01 | n.d. | 0.04 | 0.10 | 27.50 | 6.02 | n.d. | n.d. | 98.74 | 0.12 |
| MN-3 an 2 | n.d. | 1.37 | 0.03 | 2.53 | n.d. | 19.14 | 43.15 | 0.04 | n.d. | 0.04 | 0.02 | 27.84 | 6.00 | n.d. | n.d. | 100.16 | 0.07 |
| MN-3 an 3 | n.d. | 2.75 | 0.07 | 2.06 | n.d. | 17.58 | 42.48 | 0.04 | n.d. | 0.05 | 0.04 | 27.22 | 6.19 | n.d. | n.d. | 98.47 | 0.14 |
| MN-3 an 4 | n.d. | 2.42 | 0.10 | 1.88 | 0.06 | 17.76 | 42.67 | 0.04 | n.d. | n.d. | 0.02 | 27.51 | 6.41 | n.d. | 0.07 | 98.93 | 0.12 |
| MN-3 an 5 | n.d. | 2.54 | 0.06 | 1.96 | 0.02 | 17.45 | 42.80 | 0.02 | n.d. | 0.03 | 0.02 | 27.55 | 6.38 | n.d. | n.d. | 98.82 | 0.13 |
| MN-3 an 6 | n.d. | 2.60 | 0.07 | 1.98 | n.d. | 17.39 | 42.87 | 0.04 | n.d. | 0.02 | 0.08 | 27.37 | 6.11 | n.d. | n.d. | 98.51 | 0.13 |
| MN-6 an 1 | n.d. | 4.35 | 0.06 | 2.11 | n.d. | 16.51 | 42.24 | 0.01 | n.d. | n.d. | 0.03 | 27.11 | 6.43 | n.d. | n.d. | 98.86 | 0.21 |
| MN-6 an 2 | n.d. | 4.41 | 0.11 | 2.22 | n.d. | 17.01 | 41.82 | 0.04 | n.d. | 0.03 | n.d. | 27.18 | 6.17 | n.d. | 0.01 | 99.01 | 0.21 |
| MN-6 an 3 | n.d. | 4.57 | 0.20 | 2.08 | n.d. | 16.89 | 42.08 | 0.03 | n.d. | 0.01 | 0.05 | 27.44 | 6.74 | n.d. | n.d. | 100.09 | 0.21 |
| MN-2 an 1 | 0.01 | 0.49 | 0.03 | 1.52 | n.d. | 19.61 | 43.55 | 0.03 | n.d. | 0.07 | 0.08 | 27.47 | 6.28 | n.d. | n.d. | 99.13 | 0.02 |
| MN-2 an 2 | n.d. | 0.37 | 0.03 | 1.61 | n.d. | 19.43 | 43.66 | 0.04 | n.d. | 0.07 | 0.05 | 27.77 | 6.32 | n.d. | 0.03 | 99.36 | 0.02 |
| MN-2 an 3 | n.d. | 0.78 | 0.02 | 1.40 | 0.05 | 19.86 | 43.58 | 0.02 | n.d. | 0.03 | 0.01 | 27.29 | 6.60 | n.d. | n.d. | 99.65 | 0.04 |
| MN-2 an 4 | n.d. | 1.04 | 0.03 | 1.42 | n.d. | 19.64 | 42.92 | 0.06 | n.d. | 0.06 | 0.08 | 26.88 | 6.41 | n.d. | n.d. | 98.53 | 0.05 |
| MN-2 an 5 | 0.02 | 0.81 | 0.04 | 1.37 | n.d. | 20.43 | 42.28 | 0.03 | n.d. | n.d. | n.d. | 26.91 | 6.21 | n.d. | 0.01 | 98.10 | 0.04 |
| MN-4_an1 | 0.01 | 1.10 | 0.04 | 1.45 | 0.01 | 19.54 | 43.63 | n.d. | 0.78 | 0.05 | 0.13 | 27.61 | 7.12 | n.d. | n.d. | 101.48 | 0.05 |
| MN-5_an1 | n.d. | 5.46 | 1.09 | 1.84 | 0.03 | 15.87 | 42.59 | 0.02 | 0.07 | 0.05 | 0.17 | 28.24 | 6.80 | 0.02 | n.d. | 102.23 | 0.26 |
| MN-10_an1 | n.d. | 5.15 | 0.33 | 2.42 | n.d. | 16.02 | 42.76 | n.d. | 1.15 | 0.03 | 0.16 | 27.84 | 6.11 | 0.01 | n.d. | 101.99 | 0.24 |
| MN-9_an1 | n.d. | 2.61 | 0.20 | 2.18 | n.d. | 17.85 | 42.74 | n.d. | 1.35 | 0.04 | 0.14 | 27.77 | 6.46 | 0.01 | n.d. | 101.36 | 0.13 |
| MN-11_an1 | n.d. | 2.12 | 0.38 | 1.66 | n.d. | 17.06 | 42.08 | n.d. | 3.57 | 0.06 | 0.13 | 27.55 | 6.97 | 0.02 | n.d. | 101.59 | 0.11 |
| MN-11_an2 | 0.02 | 2.04 | 0.49 | 1.74 | n.d. | 16.90 | 41.79 | n.d. | 3.36 | 0.07 | 0.13 | 27.64 | 6.87 | n.d. | n.d. | 101.05 | 0.11 |
| MN-12_an1 | 0.02 | 3.62 | 0.09 | 3.73 | n.d. | 16.70 | 42.61 | n.d. | 0.76 | n.d. | 0.12 | 28.58 | 4.76 | n.d. | n.d. | 100.99 | 0.18 |
| MN-13_an1 | n.d. | 4.76 | 0.36 | 3.04 | n.d. | 15.50 | 41.76 | n.d. | 2.77 | 0.04 | 0.13 | 27.28 | 5.46 | n.d. | n.d. | 101.09 | 0.23 |
| MN-1_an1 | n.d. | 2.32 | 0.13 | 2.38 | n.d. | 16.80 | 42.77 | 0.01 | 2.95 | 0.06 | 0.13 | 28.17 | 5.83 | n.d. | n.d. | 101.54 | 0.21 |
| MN-1_an2 | n.d. | 2.38 | 0.08 | 2.46 | n.d. | 16.63 | 42.45 | n.d. | 3.20 | 0.07 | 0.14 | 27.83 | 6.01 | 0.02 | n.d. | 101.28 | 0.13 |
| MN-14_an1 | n.d. | 6.21 | 0.06 | 1.09 | n.d. | 15.59 | 43.05 | n.d. | n.d. | 0.11 | 0.16 | 27.93 | 6.90 | 0.01 | n.d. | 101.11 | 0.28 |
| MN-14_an6 | 0.02 | 6.30 | 0.04 | 1.14 | 0.01 | 15.64 | 43.29 | 0.01 | 0.05 | 0.14 | 0.17 | 28.31 | 6.74 | 0.05 | n.d. | 101.90 | 0.20 |
| **Average** | **0.00** | **2.88** | **0.16** | **1.97** | **0.01** | **17.57** | **42.71** | **0.02** | **0.77** | **0.05** | **0.09** | **27.61** | **6.32** | **0.01** | **0.01** | **100.15** | **0.14** |

n.d. = not detected.

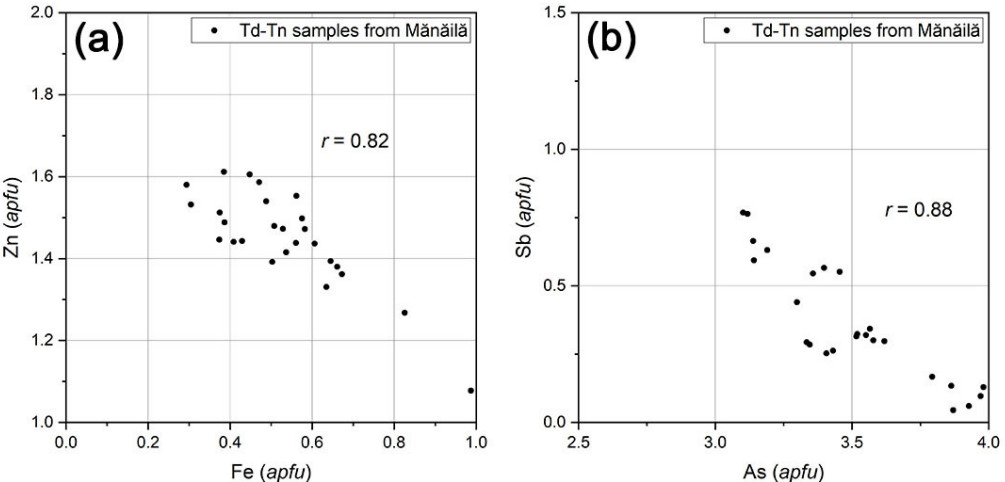

**Figure 11.** Distribution within the series of tetrahedrite-tennantite from the Mănăilă ore deposit with a correlation between Fe and Zn *apfu* (**a**); and between As and Sb *apfu* (**b**).

The high As contents correlate with sulfur concentration. Sb appears in lower amounts and its *apfu* varies between 0.04–0.77. As and Sb occupy the *D* sites in the tennantite structure. In some samples, As and Sb are substituted by Bi, but in most samples, Bi is below the detection limit. Bi occurs mostly in the copper mineralization up to 0.26 *afpu*. Bi contents in tetrahedrites were also reported by Buzatu et al. [27] in hydrothermal mineralization from Baia Sprie, Eastern Carpathian. In almost all samples, a sulfur deficit is noted because the *afpu* value is below 13, with some exceptions (Table 7).

**Table 7.** Structural formulae calculated in atoms per formula unit (*apfu*) based on 29 atoms of tennantite.

| Sample Name | Cu | Zn | Fe | Ag | As | Sb | Bi | Pb | Mn | Cd | S |
|---|---|---|---|---|---|---|---|---|---|---|---|
| MN-3 an 1 | 10.19 | 1.39 | 0.50 | 0.01 | 3.62 | 0.30 | - | 0.01 | - | 0.01 | 12.97 |
| MN-3 an 2 | 10.09 | 1.36 | 0.67 | - | 3.79 | 0.17 | - | - | 0.01 | 0.01 | 12.90 |
| MN-3 an 3 | 10.16 | 1.44 | 0.56 | 0.01 | 3.57 | 0.34 | - | - | 0.01 | 0.01 | 12.90 |
| MN-3 an 4 | 10.14 | 1.48 | 0.51 | 0.01 | 3.58 | 0.30 | - | - | 0.01 | - | 12.96 |
| MN-3 an 5 | 10.17 | 1.47 | 0.53 | 0.01 | 3.52 | 0.31 | - | - | 0.01 | - | 12.97 |
| MN-3 an 6 | 10.23 | 1.42 | 0.54 | 0.01 | 3.52 | 0.32 | - | 0.01 | 0.01 | - | 12.95 |
| MN-6 an 1 | 10.13 | 1.50 | 0.58 | 0.01 | 3.36 | 0.54 | - | - | - | - | 12.88 |
| MN-6 an 2 | 10.02 | 1.44 | 0.61 | 0.02 | 3.46 | 0.55 | - | - | 0.01 | - | 12.90 |
| MN-6 an 3 | 9.98 | 1.55 | 0.56 | 0.03 | 3.40 | 0.57 | - | - | 0.01 | - | 12.90 |
| MN-2 an 1 | 10.28 | 1.44 | 0.41 | - | 3.93 | 0.06 | - | 0.01 | 0.01 | 0.01 | 12.85 |
| MN-2 an 2 | 10.26 | 1.44 | 0.43 | - | 3.87 | 0.04 | - | - | 0.01 | 0.01 | 12.93 |
| MN-2 an 3 | 10.27 | 1.51 | 0.37 | - | 3.97 | 0.10 | - | - | 0.01 | - | 12.75 |
| MN-2 an 4 | 10.26 | 1.49 | 0.39 | - | 3.98 | 0.13 | - | 0.01 | 0.01 | 0.01 | 12.73 |
| MN-2 an 5 | 10.13 | 1.45 | 0.37 | 0.01 | 4.15 | 0.10 | - | - | 0.01 | - | 12.78 |
| MN-4 an1 | 10.17 | 1.61 | 0.38 | 0.01 | 3.86 | 0.13 | 0.06 | 0.01 | - | 0.01 | 12.76 |
| MN-5 an1 | 9.93 | 1.54 | 0.49 | 0.15 | 3.14 | 0.66 | - | 0.01 | - | 0.01 | 13.05 |
| MN-10 an1 | 10.04 | 1.39 | 0.65 | 0.05 | 3.19 | 0.63 | 0.01 | 0.01 | - | - | 12.95 |
| MN-9 an1 | 10.03 | 1.47 | 0.58 | 0.03 | 3.55 | 0.32 | 0.10 | 0.01 | - | - | 12.91 |
| MN-11 an1 | 9.98 | 1.61 | 0.45 | 0.05 | 3.43 | 0.26 | 0.26 | 0.01 | - | 0.01 | 12.95 |
| MN-11 an2 | 9.93 | 1.59 | 0.47 | 0.07 | 3.41 | 0.25 | 0.24 | 0.01 | - | 0.01 | 13.02 |
| MN-12 an1 | 9.92 | 1.08 | 0.99 | 0.01 | 3.30 | 0.44 | 0.05 | 0.01 | - | - | 13.19 |
| MN-13 an1 | 9.98 | 1.27 | 0.83 | 0.05 | 3.14 | 0.59 | 0.20 | 0.01 | - | 0.01 | 12.92 |
| MN-1_an1 | 10.04 | 1.33 | 0.63 | 0.02 | 3.35 | 0.28 | 0.21 | 0.01 | - | 0.01 | 13.11 |
| MN-1_an2 | 10.03 | 1.38 | 0.66 | 0.01 | 3.33 | 0.29 | 0.22 | 0.01 | - | 0.01 | 13.04 |
| MN-14_an1 | 10.15 | 1.58 | 0.29 | 0.01 | 3.12 | 0.76 | - | 0.01 | - | 0.01 | 13.05 |
| MN-14_an6 | 10.12 | 1.53 | 0.30 | 0.01 | 3.10 | 0.77 | - | 0.01 | - | 0.02 | 13.12 |
| **Average** | **10.10** | **1.45** | **0.53** | **0.02** | **3.52** | **0.35** | **0.05** | **0.01** | **-** | **0.01** | **12.94** |

The average stoichiometric formula of tennantites calculated on the basis of 29 atoms is: $(Cu, Ag)_{10.12}(Fe, Zn, Mn, Cd, Pb)_{1.99}(As, Sb, Bi)_4 S_{12.94}$ which is close to the ideal formula of tennantite [28]. The tennantites formula from Mănăilă is slightly variable (Table 7) because the values of As *afpu* (3.10–4.15) and Sb *afpu* (0.04–0.77) are in the same range of variation. It is obvious that the studied tennantites are homogeneous since the ratio Sb/(Sb + As) is close to zero (0.09), with a variation of 0.01–0.20, indicating a pure tennantite member.

In recent years, several Raman studies on tetrahedrite-tennantite solid solution series have pointed out systematic changes in terms of peak position and changes of relative intensity [25,29]. The fundamental modes ($\nu_1$ to $\nu_4$) of As-rich members of the series are clearly observed in the Raman spectra (Figure 12). The most notable spectral feature is represented by the $\nu_1$ mode located at 385 cm$^{-1}$ which can be assigned to the symmetric stretching of pyramidal groups AsS$_3$. The other fundamental modes are displayed as shoulders, as follows: antisymmetric stretching $\nu_3$ at 365 cm$^{-1}$; symmetric bending $\nu_2$ at 342 cm$^{-1}$; and antisymmetric bending $\nu_4$ at 312 cm$^{-1}$. The lattice modes are observed in the low wavenumber region of the Raman spectra (below 202 cm$^{-1}$). Considering the previous Raman studies [25,29], the spectral features are in agreement with the chemistry (Tables 6 and 7).

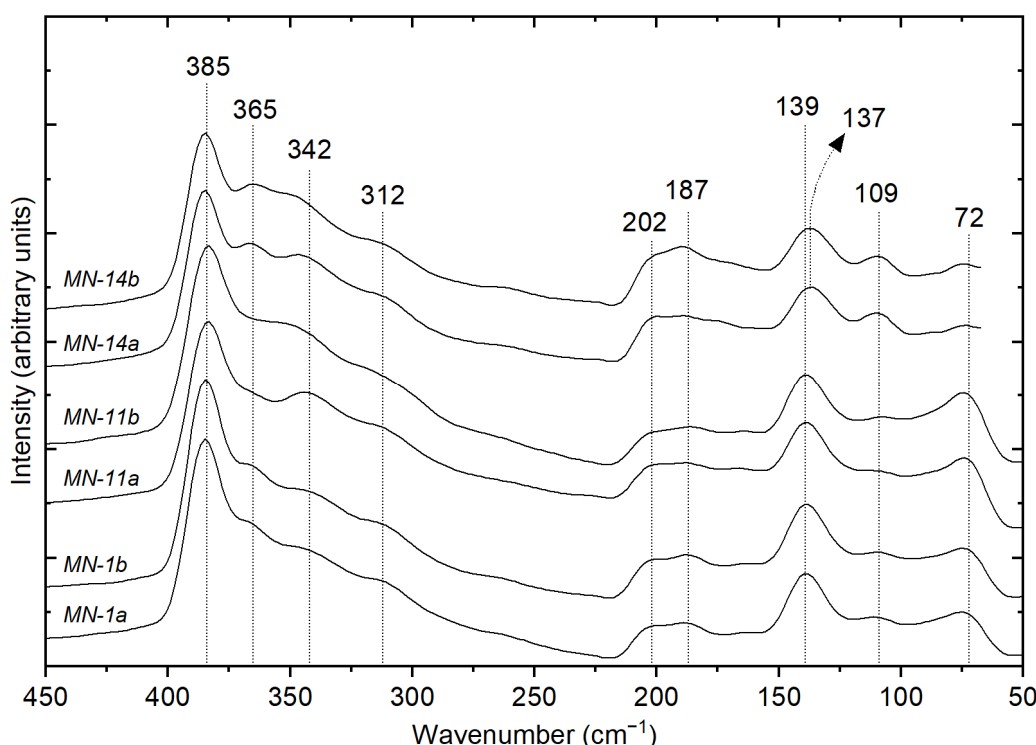

**Figure 12.** Raman spectra of the tennantites from the Mănăilă ore deposit. The spectra are displayed vertically for clarity.

## 5. Discussion

### 5.1. Ore Mineral Formation

The sulfide mineralizations from Mănăilă are unique within the VMS deposits of the Eastern Carpathians. The mineralogical composition is very different from the other deposits, including those in the Fundu Moldovei area. Regarding the accumulation genesis of syngenetic pyrites, it is accepted by several authors as being of volcanogenic–sedimentary type, regionally metamorphosed in the facies of greenschists. Moreover, this hypothesis is also supported by the association of massive pyrite accumulations with rocks that initially had an acid character (metariolites, acid metatuffs) and the appearance of schistose and cataclastic textures [7,16,30]. Some gangue minerals with which the ore is associated (e.g., quartz) show preferential orientation, the quartz being synmetamorphically recrystallized [30]. Sulfides show fracturing followed by recrystallization [30].

These mineralizations belong to a metallogenetic province in the Eastern Carpathian and were considered syngenetic of volcanogenic–sedimentary origin. They are spatially associated with a petrogenetic alignment with Cambrian rhyolitic volcanism during periods of volcanic calm [31] and later regionally metamorphosed. The horizon of massive sulfides from Mănăilă is associated with rhyolitic metatuffs. Although these mineralizations could be attributed to the Kuroco type, it is highly challenging to include them in models of global tectonics based on the petrographic and mineralogical similarities.

There are several models that have tried to explain the formation of sulfides through sedimentation processes around the submarine source [7,31]. In the rhyolitic metavolcanites and the chlorite schists located below the ore, irregular and discontinuous pyrite disseminations frequently occur [31].

Depositional models are based on the existence of some hydrothermal convective cells with seawater which transported the dissolved metal load from the crust to the surface. Rhyolitic volcanism contributed through a local elevation of the gradient, generating favorable conditions for the installation of convective cells. The precipitation of sulfides from the hydrothermal solutions around the emission centers probably took place in areas

with negative relief where the sedimentation of the precipitate was not disturbed by the agitation of the marine environment [31].

The regionally metamorphosed character of the pyrite-polymetallic, pyrite copper, compact, and precompact copper ores from Mănăilă is also denoted by their structure, composition, and alignment with the metamorphic formations. The mineralization association with rhyolitic metatuffs and sericite-chlorite schists with albite porphyroblasts would indicate the volcanogenic source of the sulfides. The metamorphic character is not observed in the precompact quartz-copper mineralizations with abundant chalcopyrite, bornite, tetrahedrite with bismuth and other minerals: bismuth sulfosalt, unnamed mineral, mawsonite, quartz, barite, and calcite. This mineralization appears in the form of bands at the upper part of the pyrite-polymetallic and pyrite copper compact mineralization, and we consider it epigenetic.

The formation of this paragenesis can be more difficult to explain by the metamorphosed volcanogenic character. This mineralization is completely different from those of the VMS type and would be of a different origin. If we consider metamorphism as a petrogenetic process on the same level as magmatism and sedimentogenesis, then we must treat such paragenesis as a result of metamorphism. The presence of retromorphism in the sulfide schists of the Tulgheș series in the Fundu Moldovei area would suggest the formation of a paragenesis with sulfides and sulfosalts of copper and bismuth. Balintoni and Chițimuș [16] reported certain polymorphism processes with a final retromorphism character. Gradual transitions from retromorphosed or early retromorphosed mesometamorphic rocks to porphyrogenic rocks, to sericite-chlorite schists, and finally to chlorite-quartz schists or even only chlorite schists were reported by Popescu [32] in the massive pyrites area from Bălan; the latter being closely associated with ore deposits of massive pyrites.

Some mineralizations, such as those of Mestecăniș and Delnița, are found in the Fundu Moldovei area, arranged on directional structural alignments (reverse faults related to the nappe systems) for which no regionally metamorphosed volcanogenic-sedimentary genesis has been assigned. These were considered epigenetic, being related to hidden magmatic manifestations, either regenerated or formed in connection with transformed fractures [33,34], which allowed access to the surface of metasomatic hydrothermal solutions. These mineralizations have a discordant character and appear as lenses, disseminations, and veins in sericite-chlorite schists, and are considered epigenetic of the Mesozoic (Triassic) age [34]. The mineralizations from Mănăilă are on the same alignment but much further north of Mestecăniș. Some metasomatic hydrothermal solutions had access to the fractures in the area, depositing and remobilizing some of the metals from the regionally metamorphosed volcanogenic-sedimentary mineralization. Such metasomatic or retromorphic hydrothermal solutions would have generated the mineralization from the precompact quartz-copper paragenesis with large amounts of copper minerals, including the newly identified phase (unnamed mineral).

### 5.2. Mineralogical Comparison with Other Deposits

VMS deposits from Eastern Carpathian including the mineralization from the Fundu Moldovei area have a simple mineralogical composition that includes pyrite, chalcopyrite, sphalerite, galena, and small amounts of pyrrhotite, arsenopyrite, and tetrahedrite. The precompact quartz-copper ore with abundant chalcopyrite, bornite, and tetrahedrite with bismuth and other minerals: bismuth sulfosalt, unnamed mineral, mawsonite, quartz, barite, and calcite is different from the other VMS deposits in the Fundu Moldovei area.

The unnamed mineral is similar to the chemistry of "cuproplumbite" [35] with the formula $Cu_{10}PbS_6$; other times, it was described with the formula $Cu_6PbS_4$ as alisonite [36]. Alisonite has been reported at Grande Mine (Chile) and Daly-Judge Mine (Utah, USA). The validity of this mineral was also questioned by Dana [35] who considered it a mixture. Alisonite was also described in the Deposits of Ducktown, Tennessee by Emmons and Laney [37], but they consider it to be a lead-rich variety of covellite. Kucha and Salamon [38] described a mineral similar to the unnamed mineral from the present study, and with the

formula $Cu_{21}PbS_{16}$, in association with chalcocite in the copper mineralizations of the Lublin area. Later, Shihui et al. [39] described an unnamed sulfide mineral with the formula $CuPbS_2$. This last mineral has a different composition from the previous ones because the atoms per formula unit (*apfu*) of the metals is equal to that of the metalloids. Because the *apfu* for metals is greater than metalloids in the empirical formula, the unidentified mineral from Mănăilă is more similar to the one described by Kucha and Salamon [38] than to the one reported by Shihui et al. [39] and its structural formula would be the following: $Cu_{21}Fe(Pb, Ag)_2S_{14}$ or $(Cu, Fe)_{11}(Pb, Ag)S_7$. Nevertheless, the mineral differs from that characterized as alisonite as well as the one described by Kucha and Salamon [38] and could be a new mineral if the structural data can be obtained.

The Raman spectrum of the unnamed mineral is typical for sulfosalt structures where the vibrational modes appear at lower wavenumbers [25,29] but are different from any similar sulfides or sulfosalts. Bournonite shows two intense Raman bands at 296 and 325 cm$^{-1}$ [25], the chalcocite spectrum is dominated by the 213–216 cm$^{-1}$ lines with several lines in the higher region, and the covellite main peak is observed at 469 cm$^{-1}$ [40].

Mawsonite was identified for the first time in Romania. The chemical analysis (Table 4) is identical to that of mawsonite reported by Markham and Lawrence [23]. This study presents for the first time the Raman spectrum of mawsonite, with no other discussions being reported so far in previous literature. One infrared study was mentioned by Chiad et al. [41] where the authors assign the IR vibrational modes between 400 and 1500 cm$^{-1}$ to Sn-S bonds (560 and 1384 cm$^{-1}$), Cu-S modes (1117 cm$^{-1}$), and the Fe-S stretching mode (473 cm$^{-1}$). The mawsonite has a sphalerite-type superstructure with distorted tetrahedra of SnS$_4$ and CuS$_4$ that share corners, stacked along the same axis with FeS$_4$ edges-sharing tetrahedra [42].

According to the latest classifications [28], the tetrahedrite from Mănăilă is tennantite-(Zn) since it contains predominantly As, and Zn is in greater quantity than Fe (Tables 6 and 7). Copper varies between 9.93—10.29 *apfu* and can appear as Cu$^+$ in *A* and *B* structural sites and Cu$^{2+}$ in the *C* site [28]. Cu$^{2+}$ especially appears in samples with a Cu content > 10 *apfu* [43]. Divalent copper (*apfu*) was calculated according to Apopei [44] by using the following relation: Cu$^{2+}$ = Cu$_{total}$ + Ag—10, while Cu+ was obtained as Cu$^+$ = Cu$_{total}$—Cu$^{2+}$. By adding the Cu$^{2+}$ *apfu* to Zn+Fe *apfu* (Table 7) it was found that the 12 *apfu* for metal cations in the tennantite structure are exceeded, due to an excess of Cu$^{2+}$ or $Me^{2+}$ (divalent metals) [28].

The presence of bornite, chalcocite, tetrahedrite with bismuth and bismuth and tin sulfosalts, especially in precompact copper ores from Mănăilă, is not typical of VMS deposits from Eastern Carpathians. Barite is identified in the mineralizations from Ostra [42] and is considered epigenetic. The strong brecciation of pyrite by the bornite-chalcocite association deposited later would indicate the formation of the quartz-precompact copper ore from metasomatic-type epigenetic solutions. This is an argument to consider the precompact quartz-copper ore from Mănăilă as epigenetic.

## 6. Conclusions

The electron microprobe analyses on Mănăilă samples provide important information about the ore-forming conditions and chemical variations within the mineralogy of the deposit.

The Mănăilă mineralization is concentrated in the rocks of the Tulgheș Lithogroup in the crystalline-Mesozoic zone of the Eastern Carpathians. The polymetallic sulfide mineralizations from Manăilă are part of the polymetallic mineralization belt associated with the epimetamorphic schists of the Tulgheș Lithogroup in the Bucovinian nappe (Eastern Carpathians). The mineralizations from Mănăilă appear in the form of a stratiform lens compartmentalized according to the direction and inclination of the faults. The precompact quartz-copper mineralization from Mănăilă is an epigenetic type ore and is different from pyrite-polymetallic, pyrite copper, and compact considered VMS types.

Using optical and chemical determinations, the following types of ore were identified: pyrite-polymetallic, pyrite copper, compact and precompact copper, and quartz-precompact copper. Compact and precompact copper mineralization and precompact quartz-copper mineralization are characterized by a different mineralogical composition than other polymetallic deposits in the Tulgheș Lithogroup. It is mainly composed of copper minerals: chalcopyrite, bornite, chalcocite, tetrahedrite, mawsonite, wittichenite, an unnamed mineral with the chemical formula $(Cu, Fe)_{11}(Pb, Ag)S_7$, with quartz and barite as gangue minerals. We consider this mineral paragenesis to be different from the regionally metamorphosed volcanic-sedimentary type. This paragenesis could be formed by retromorphism or by the circulation of metasomatic hydrothermal solutions that had access to the major fractures in the investigated area.

Raman spectra of tennantite are in good agreement with those reported previously in the literature and show a strong correlation between spectroscopic parameters and the Sb/(Sb + As) content ratio. The Raman spectrum of the unnamed mineral is typical for sulfosalt structures, with vibrational modes at lower wave numbers, but different from any similar sulfides or sulfosalts. New Raman spectra of natural mawsonite are presented for the first time in this study.

The precompact quartz-copper mineralization differs from the ones of Tulgheș Lithogroup by large amounts of bornites and chalcocites described for the first time.

Tennantite occurs in much larger quantities than in other occurrences from the Tulgheș Lithogroup. The low content of bismuth in tennantite is pointed out for the first time. Wittichenite was mentioned for the first time in the hosted mineralizations in the Tulgheș Lithogroup through quantitative and not descriptive data.

Mawsonite represents the first occurrence identified so far on the territory of Romania.

The unnamed mineral is a cuproplumbite with the formula $(Cu, Fe)_{11}(Pb, Ag)S_7$, identified for the first time and included in the sulfides group. It differs from the other cuproplumbites described so far by its chemical composition, and additional structural data are required.

**Author Contributions:** Conceptualization, G.D.; methodology, G.D., A.I.A., A.B., A.E.M. and F.D.; software, A.I.A. and A.B.; validation, G.D., F.D., A.E.M., A.B. and A.I.A.; formal analysis, G.D., A.B., A.I.A., F.D. and A.E.M.; investigation, G.D., A.B. and A.I.A.; resources, G.D. and F.D.; writing—original draft preparation, G.D. and F.D.; writing—review and editing, G.D., F.D., A.B., A.I.A. and A.E.M.; visualization, A.B. and A.I.A.; supervision, G.D.; project administration, G.D. All authors have read and agreed to the published version of the manuscript.

**Funding:** This research received no external funding.

**Data Availability Statement:** Not applicable.

**Acknowledgments:** This work was supported by the Carpathian Association of Environment and Earth Sciences, Baia Mare, Romania. Thanks are also extended to the staff of electron microanalysis and Raman laboratory (from the Earth Science Institute, Slovak Academy of Sciences, Banská Bystrica, Slovakia): Mikuš, T. and Milovská, S. We also extend our gratitude to the anonymous reviewers for their time spent reading the first draft of our manuscript and for their valuable suggestions.

**Conflicts of Interest:** The authors declare no conflict of interest.

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
