# Peer review of "New Mineral Occurrences in Massive Sulfide Deposits from Mănăilă, Eastern Carpathians, Romania"

_minerals, doi:10.3390/min13010111_

Round 1
Reviewer 1 Report
Dear Colleague,
I have examined your manuscript entitled "New Mineral Occurrences in Massive Pyrite Ore Deposits from Mănăilă, Eastern Carpathians, Romania". I have a lot of questions, especially in the presentation of materials and the discussion part of the Manuscript. Therefore, I have concluded that the manuscript is suitable for publication in the journal after applying major revisions. I hope my comments would be useful for improving the manuscript.

Author Response
Dear Editor and Reviewers,
Herewith, we are submitting the revised version of the manuscript entitled “New Mineral Occurrences in Massive Sulfide Deposits from Mănăilă, Eastern Carpathians, Romania”, manuscript no: minerals-2107915, to be considered for publication in Minerals journal.
We have carefully considered the comments of the two reviewers. We are grateful for their time spent reviewing this manuscript and for the valuable comments and suggestions made to improve the paper.
We addressed each specific comment pointed out by reviewer 1 in the attached file.
With kind regards,
The Authors.

Reviewer 2 Report
Dear Authors,
Manuscript minerals-2107915 may be published after some improvements of the manuscript as follows:
1. At section 3 add information on the optical microscopy measurements.
2. Increase figure and text size in Figure 1 (it is illegible). Modify A,B,C to a, b,c. Eliminate D of caption. Figure 1 and 2 should be independent.
3. Organize the Results section 4.2. in subsections, dedicated to different mineral investigations. Also Results and Conclusions sections would benefit of using bullet points emphasizing the main discoveries.
4.Add scale bars to images of Fig. 3, 4, 5, 7, and 9.
5.Increase picture resolution of Fig 10, increase picture and text size.
6.Revise English, some phrases are too long, check topic.
Author Response
Dear Editor and Reviewers,
Herewith, we are submitting the revised version of the manuscript entitled “New Mineral Occurrences in Massive Sulfide Deposits from Mănăilă, Eastern Carpathians, Romania”, manuscript no: minerals-2107915, to be considered for publication in Minerals journal.
We have carefully considered the comments of the two reviewers. We are grateful for their time spent reviewing this manuscript and for the valuable comments and suggestions made to improve the paper.
We addressed each specific comment pointed out by reviewer 2 in the attached file.
With kind regards,
The Authors.

Round 2
Reviewer 1 Report
Many of the revisions from the reviewer comments were made, and the manuscript is overall improved. The added figures are mostly appropriate. However, there are still moderate revisions necessary as indicated below.
1. Terminology of structural types of ores used by the authors is not typical for massive sulfide deposits. Why? Massive, sulfide breccia, brecciated, layered, banded, and other structures is more appropriate for VMS ores.
2. Some photographs are not convincing of interpretations given in the captions to figures. Not all of these additions satisfy my comment. It is necessary to show the relationships between minerals (intergrowth, replacement, fouling, filling of pore space, fine grained pyrite with interstitial…, pyrite cemented by …, etc.). Help the reader figure out what is important in these photos (Figs. 4-8). Sometimes long detailed figure captions go a long way to pointing out specific relationships.
3. Do not use that pyrite is a metallic mineral. In VMS literature, pyrite is most common ore mineral.
4. Unfortunately, the authors did not separate the research results from the discussion. My opinion remains unchanged. In the manuscript, only the data obtained by the authors can be presented as results, and all the arguments of the authors up to the discovery of minerals and comparison of their composition with early studies should be in the discussion section.
Author Response
Dear Editor and Reviewer,
Herewith, we are submitting the revised version #2 of the manuscript entitled “New Mineral Occurrences in Massive Sulfide Deposits from Mănăilă, Eastern Carpathians, Romania”, manuscript no: minerals-2107915, to be considered for publication in Minerals journal.
We have carefully considered the comments of the first reviewer. We are grateful for the time spent reviewing this manuscript and for the valuable comments and suggestions made to improve the paper.
We will address further each specific comment pointed out by the reviewer.
With kind regards,
-- The Authors.
